# Vnn1 pantetheinase limits the Warburg effect and sarcoma growth by rescuing mitochondrial activity

Caroline Giessner[1],*, Virginie Millet[1],*, Konrad J Mostert[2], Thomas Gensollen[1], Thien-Phong Vu Manh[1], Marc Garibal[3] , Binta Dieme[3], Noudjoud Attaf-Bouabdallah[1], Lionel Chasson[1], Nicolas Brouilly[4], Caroline Laprie[5], Tom Lesluyes[6], Jean Yves Blay[6], Laetitia Shintu[7] , Jean Charles Martin[3], Erick Strauss[2] , Franck Galland[1], Philippe Naquet[1]

**Like other tumors, aggressive soft tissue sarcomas (STS) use glycolysis rather than mitochondrial oxidative phosphorylation (OXPHOS) for growth. Given the importance of the cofactor co-enzyme A (CoA) in energy metabolism, we investigated the impact of Vnn1 pantetheinase—an enzyme that degrades pantetheine into pantothenate (vitamin B5, the CoA biosynthetic precursor) and cysyteamine—on tumor growth. Using two models, we show that Vnn1+ STS remain differentiated and grow slowly, and that in patients a detectable level of VNN1 expression in STS is associated with an improved prognosis. Increasing pantetheinase activity in aggressive tumors limits their growth. Using combined approaches, we demonstrate that Vnn1 permits restoration of CoA pools, thereby maintaining OXPHOS. The simultaneous production of cysteamine limits glycolysis and release of lactate, resulting in a partial inhibition of STS growth in vitro and in vivo. We propose that the Warburg effect observed in aggressive STS is reversed by induction of Vnn1 pantetheinase and the rewiring of cellular energy metabolism by its products.**

## Introduction

Tumor development requires a constant adaptation to environmental stress and involves metabolic rewiring (Fouad & Aanei, 2017). Many tumors switch to aerobic glycolysis (the Warburg effect) to optimize cell fitness and growth under harsh and competitive growth conditions (Vander Heiden, Cantley et al, 2009). Similarly, mitochondrial metabolites can be partly reoriented from energy production towards anabolism (Mullen et al, 2012). Uncoupling of the tricarboxylic acid cycle from OXPHOS limits

reactive oxygen species production while preserving an efficient glycolytic flow. By limiting glucose oxidation, the Warburg effect contributes to anoikis resistance and metastasis (Lu et al, 2015). These metabolic transitions provoke the release of metabolites that are exchanged between the tumor and cells in its environment. Lactate produced by glycolytic cells provides carbon to respiring tumor cells (Lisanti et al, 2013) and modulates macrophage polarization (Colegio et al, 2014). Competition for energy resources between nontumor and tumor cells is also exploited by cancers to escape immune responses (Chang et al, 2015). Because cancerous cells can become addicted to new energy resources, targeting these indirect oncogenic pathways is becoming an attractive antitumor strategy (Galluzzi et al, 2013). In addition, identification of regulators of metabolic transitions should provide biomarkers of tumor fitness and new metabolites with potential therapeutic interest.

We identified the Vnn1 pantetheinase activity as a key regulator of tissue tolerance to stress in various diseases (Pitari et al, 2000; Naquet et al, 2014). This stress-inducible enzyme hydrolyzes pantetheine (PantSH) (Maras et al, 1999)—a degradative product of coenzyme A (CoA)—into pantothenate (vitamin B5) and cysteamine, a small aminothiol which posttranslationally modifies target proteins and impacts signaling (Naquet et al, 2014). Although this enzyme activity plays a limited role under homeostatic conditions, its induction reflects a local adaptation to metabolic or oxidative stress as shown in the gut (Martin et al, 2004; Berruyer et al, 2006), liver (Rommelaere, Millet et al, 2013b; van Diepen et al, 2014; Ferreira et al, 2016), and connective tissue (Dammanahalli et al, 2012), where it participates in healing processes.

Cancers are often viewed as never-healing wounds (Dvorak, 1986) and we anticipated that Vnn1 expression might be induced in some tumors and impact their progression through metabolic rewiring. To test this hypothesis, we introduced the *vnn1* gene deletion in p16[Ink4A]/ p19[Arf]-deficient mice (p16/19[−/−]) which spontaneously develop

---

[1]Aix Marseille Univ, Centre National de la Recherche Scientifique, Institut National de la Santé et de la Recherche Médicale, Centre d'Immunologie de Marseille Luminy, Marseille, France   [2]Department of Biochemistry, Stellenbosch University, Stellenbosch, South Africa   [3]Aix Marseille Univ, Institut National de la Santé et de la Recherche Médicale, Institut National de la Recherche Agronomique, C2VN, Marseille, France   [4]Aix Marseille Univ, Centre National de la Recherche Scientifique, Institut National de la Santé et de la Recherche Médicale, Institut de Biologie de Développement de Marseille, Marseille, France   [5]Laboratoire Vet-Histo, Marseille, France   [6]Centre Lyon Bérard, Université Claude Bernard, Lyon 1, Lyon Recherche Innovation contre le Cancer, Lyon, France   [7]Aix Marseille Université, Centre National de la Recherche Scientifique, Centrale Marseille, ISM2, Marseille, France

Correspondence: naquet@ciml.univ-mrs.fr; estrauss@sun.ac.za
*Caroline Giessner and Virginie Millet contributed equally to this work.

various tumor types, including lymphomas, sarcomas, and carcinomas (Sharpless et al, 2002) and probed whether Vnn1 deficiency impacted the development of specific cancers and modulated their metabolic environment. By combining new genetic models with transcriptomic, metabolomic, and electron microscopy approaches, we identified the Vnn1 pantetheinase as a tumor suppressor for the development of aggressive soft tissue sarcomas (STS). Our results provide the first experimental evidence that the two products of pantetheinase activity exert complementary effects on the tumor, explaining the basis for Vnn1's suppression on tumor growth. Specifically, we demonstrate that pantothenate production restores CoA levels leading to an enhanced mitochondrial activity, whereas cysteamine reduces glycolysis and prevents the transition to the Warburg effect, a feature of the most aggressive tumors (Casazza et al, 2014). Taken together, these results point to Vnn1 as a novel biomarker for STS prognosis, and it suggests a new route to regulate tumor growth rates through the modulation of CoA and cysteamine levels.

# Results

### Vnn1 deficiency favors soft tissue sarcoma development in p16/p19$^{-/-}$ mice

To test the contribution of Vnn1 to spontaneous tumor development, we compared mouse survival and tumor incidence in three independent cohorts of p16/p19/Vnn1$^{-/-}$ versus p16/p19$^{-/-}$ mice, derived from two independently derived crosses between p16/p19$^{-/-}$ and Vnn1$^{-/-}$ mice. As shown in Fig 1A, whereas 35% p16p19$^{-/-}$ mice progressively developed lethal tumors within 220 d, 70% p16p19/Vnn1$^{-/-}$ died of aggressive tumors before 220 d ($P$ = 0.032). Based on the survival curves, we scored the presence of tumors at the date of sacrifice by macroscopic dissection and systematic anatomopathology of harvested organs for microscopic examination (200 d or earlier when premature death occurred). Results compiled in Table 1 indicate that 53% p16p19$^{-/-}$ and 65% p16p19/Vnn1$^{-/-}$ of the mice had developed tumors at autopsy. Whereas p16p19$^{-/-}$ mice developed various tumor types with a majority of lymphomas, p16/p19/Vnn1$^{-/-}$ mice predominantly developed skin STS typed as fibrosarcomas (Fig 1B). An anatomopathology analysis scored the degree of disorganization and anisonucleosis, the mitotic and necrotic indices and classified all STS tumors from differentiated (grade I) to undifferentiated (grade III). The differentiation grade of the available subcutaneous tumors was further investigated by quantifying the expression levels of collagen 1 and α-smooth muscle actin transcripts (Fig 1C) which are conventional markers of mesenchymal cell differentiation. STS developing on the Vnn1$^{-/-}$ background were mostly grade II and III sarcomas. In the only case of skin STS (grade III) observed in Vnn1$^{+/+}$ mice, a low level of Vnn1 transcript was detected by qRT–PCR (not shown), indicating that few cells express Vnn1 in the tumor mass. Of the three cohorts of mice tested, the few other sarcomas were only found in the liver or spleen in which Vnn1 can be expressed by various cell types. These results suggest that Vnn1 can be expressed in STS and may delay their development and/or growth. In favor of this hypothesis, an analysis of VNN1 transcriptional profile in a large array of human STS gathered in the Conticabase (Chibon et al, 2010) showed that undetectable level

of VNN1 expression (observed in 198 of 349 sarcomas with complex genomics, 57%) is associated with increased risk of metastatic relapse in patients (Fig 1D).

### A novel mouse STS model

Given the slow rate of tumor development and the rare emergence of STS in Vnn1$^{+/+}$ mice, we developed new transplantable STS models derived from primary myofibroblasts, previously shown to express Vnn1 in the context of an ischemic stress (Dammanahalli et al, 2012). To bypass the senescence checkpoint, we derived myofibroblast lines obtained from p16/p19/Vnn1$^{-/-}$ and p16/p19$^{-/-}$ mice and selected those expressing comparable levels of type 1 collagen and α-smooth muscle actin (not shown). As shown in Fig S1A, cell lines obtained from a Vnn1$^{+}$ mouse express low Vnn1 levels in culture, and this expression is augmented by pharmacological agents provoking a cytosolic (transfected poly(I:C), ER [thapsigargin] or mitochondrial [antimycin A] stress). Although we could indefinitely expand these cell lines in vitro, they rarely developed into tumors after grafting them in immunodeficient nude mice. We therefore transformed the p16/p19/Vnn1$^{-/-}$ lines with an oncogenic Ras, coexpressed or not with mock (R), intact (VR), or catalytically deficient (VdR) Vnn1 (Fig S1B–D). Three independent myofibroblast tumor lines (J2A, H1, or I1) were obtained which express comparable levels of RasV12-GFP chimeric protein (R) coupled or not to wild-type (VR) or catalytically deficient (VdR) Vnn1 protein at the cell membrane (Fig S1C). Only control VR cells expressed a detectable pantetheinase activity (Fig S1D). In vitro, these cell lines showed comparable multiplication rates over several passages after being repeatedly seeded at the same density (Fig S1E).

### Vnn1 expressing tumors grow slowly and remain differentiated

As shown in Fig 2A, a subcutaneous graft of R and VdR cells in nude mice led to the development of aggressive tumors, whereas VR cells grew poorly in vivo ($P$ < 10$^{-3}$). These results were confirmed in several independent experiments with all cell lines (Fig S2A and B), and also after grafting them in immunocompetent mice (Fig 2B). This shows that expression of an enzymatically active Vnn1 pantetheinase limits their growth in vivo. To better focus our analysis on tumor cells, we performed all subsequent experiments in the nude mouse model.

To investigate the mechanisms underlying the tumor suppressive effect of Vnn1 on STS, we performed unbiased metabolomics and transcriptomic analyses of tumors. Metabolomics analysis combining liquid chromatography-mass spectrometry (LC-MS) and nuclear magnetic resonance (NMR) approaches segregated R from VR tumors (Fig 2C and D) and highlighted metabolic pathways associated with cell growth in R tumors. Some among these were recently shown to be a source of biomarkers for cancer progression (Mimmi et al, 2013; Buzzato et al, 2017; Di Virgilio & Adinolfi, 2017; Tan et al, 2017). NMR analysis further identified the presence of excess lactate and saturated/unsaturated fatty acids in R tumors, whereas VR tumors were enriched in glucose and glutathione (Fig 2D), VdR tumors showing an R tumor–like profile (Fig S2C and D). This relative enrichment in glucose versus fatty acids is suggestive of a differential consumption between R and VR tumors. The LC-MS analysis allows the detection of a larger number of metabolites grouped in

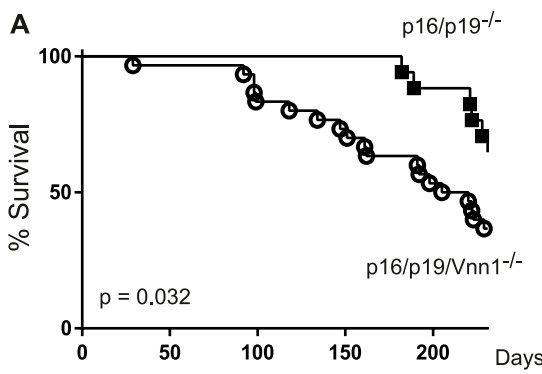

**Figure 1. Lack of Vnn1 is associated with the development of aggressive skin STS.**
**(A)** Survival curves of p16/p19/Vnn1$^{-/-}$ (n = 30) versus p16/p19$^{-/-}$ (n = 17) mice (Log rank test *P* = 0.032). **(B)** Anatomopathological typing after hematoxylin/eosin staining of tumor sections and distribution of spontaneously grown tumors in mice sacrificed at 200 d; photos represent the most frequent tumor types observed in p16/p19$^{-/-}$ (lymphoma) and p16/p19/Vnn1$^{-/-}$ (STS) tumors (magnification 200× and 400×). **(C)** Quantification of collagen I and α-smooth muscle actin transcripts by qRT–PCR analysis of available subcutaneous Vnn1$^{-/-}$ or Vnn1$^{+/+}$ STS tumors graded by histology (fibrosarcomas from grade FI–FIII). **(D)** Distribution of metastasis-free survival in patients grouped according to the relative level of VNN1 transcript abundance in the sarcoma database.

**Table 1. Tumors grown in the p16/19$^{-/-}$ mouse model.**

| Genotype | Mice with tumors at 200 d | Incidence (%) |
|---|---|---|
| p16/p19$^{-/-}$, n = 30 | n = 16 | 53 |
| | Lymphoma (spleen and liver) | 37 |
| | Histiocytic sarcoma (spleen, liver, and skin) | 31 |
| | Fibrosarcoma (spleen and skin) | 13 |
| | Sarcoma (muscle) | 6 |
| | Hemangioarcoma (spleen) | 13 |
| | Carcinoma | 0 |
| p16/p19/Vnn1$^{-/-}$, n = 32 | n = 21 | 65 |
| | Fibrosarcoma (skin) | 47 |
| | Lymphoma (spleen and liver) | 19 |
| | Carcinoma (skin and intestinal) | 10 |
| | Histiocytic sarcoma (liver) | 14 |
| | Hemangioarcoma (liver) | 5 |
| | Undifferentiated sarcoma (skin) | 5 |

pathways (i.e., choline or nucleotides) (Fig 2C and Tables S2 and S3). This analysis indicates that various pathways are differentially affected by Vnn1 expression on VR versus R tumors and is consistent with R tumors displaying features of fast growth (nucleotides, choline pathway), glucose consumption, and reduced fatty acid disposal.

To link the metabolic shifts to transcriptional programs induced in the tumor mass, we performed a transcriptional profiling and high-throughput gene set enrichment analysis (GSEA). This analysis distinguished R or VdR from VR tumors (Fig 2E). Five generic signatures that were enriched in gene sets related to growth emerged from the GSEA analysis: (mTORC1), metabolism (glycolysis, cholesterol biosynthesis, and fatty acid metabolism), hypoxia, tissue organization (collagens), and inflammation (complement, inflammatory responses, IL-6, and IFNγ signaling). The heat maps of selected leading edges are shown in Fig S3. R tumors highlighted signatures associated with intense proliferation such as glycolysis, mTORC1, and cholesterol biosynthesis pathways. In contrast, VR tumors showed a significant enrichment in genes associated with mesenchymal cell differentiation such as collagen production, a phenotype reminiscent of primary tumors observed in p16p19$^{-/-}$ mice. The latter was further documented by immunohistochemistry analysis (Fig 2F) and qRT–PCR quantification of collagen 1 and caveolin 1 transcripts using RNA extracted from total tumor mass (Fig 2G). This analysis confirmed that VR tumors express higher levels of collagen and caveolin than R tumors. Therefore, expression of Vnn1 by myofibroblast STS is associated with a differentiated status and slow growth as in the case of spontaneous tumors in the p16p19$^{-/-}$ mouse model.

### Vnn1 expression antagonizes the glycolytic switch in Ras-driven tumors

The predominant transcriptional signature associated with R tumors is linked to glycolysis and hypoxia, a hallmark of the development

of the Warburg effect. Because the transcriptomic analysis was performed on RNA from total tumor extracts, we performed a comparative qRT–PCR analysis on enriched tumor cells, after exclusion of the infiltrating CD45$^+$ hematopoietic cells known to undergo metabolic changes that could alter the interpretation of the results. As shown in Fig 3A, the expression of transcripts associated with hypoxic/glycolytic signatures (Glut1, Pdk1, Hk2, Adm, Bnip3, and Car9) was significantly enhanced in dissociated tumors ($P < 10^{-4}$). Glycolytic tumors are generally associated with fast growth. Because cysteamine and pantothenate are the two products of pantetheinase activity, they were added to in vitro cultures to test whether the Vnn1-mediated tumor suppressor effect could be recapitulated by its products alone or in combination. As shown in Fig 3B, cysteamine reduced the growth of R cell lines in vitro by 50%. This effect was not modified by adding pantothenate at different concentrations to the culture medium. Importantly, in vivo administration of cysteamine strongly reduced tumor size (Figs 3C and S4A) with partial (Glut1, Pdk1, and Hk2) modification of their hypoxic transcriptional signature (Fig S4B) but without affecting their differentiation status (Fig S4C). Increased glycolytic behavior is typically associated with lactate production, a hallmark of the Warburg effect. As shown in Fig 3D, R but not VR tumors produced high levels of lactate, and cysteamine administration to mice lowered lactate production by tumors. This finding was confirmed in vitro by Seahorse analysis where cysteamine partially reduced the glycolytic capacity of R lines in vitro (Fig 3E). Therefore, R tumors display the features of highly glycolytic, growth-oriented tumors, whereas Vnn1 exerts an anti-Warburg effect, in part through the cysteamine-mediated inhibition of glycolysis and lactate production.

### Vnn1 pantetheinase raises CoA levels and improves mitochondrial fitness in tumors

Pantothenate is the biosynthetic precursor of CoA, a cofactor that is essential for mitochondrial activity. Detection of pantothenate concentrations in tissues were at the limit of sensitivity of detection using LC-MS analysis. We therefore obtained only partial information showing that some VR but not R or cysteamine-treated R tumors showed detectable pantothenate content, although the difference was not statistically significant (Fig 4A). In contrast, CoA levels quantified by HPLC were significantly elevated in VR tumors compared with R or cysteamine-treated R tumors (Fig 4B), demonstrating for the first time that Vnn1 pantetheinase activity affects CoA levels in vivo. Cysteamine treatment had only a marginal effect on CoA concentrations, possibly as an indirect consequence of its effect on tumor growth.

We next investigated mitochondrial homeostasis. Quantification of mitochondrial content by Tom20 staining on tumor sections and MitoTracker analysis on in vitro–grown cells did not document significant differences (Fig S5A and B). We then performed a comparative EM analysis of cultured R and VR cell lines and their corresponding in vivo–grown tumors. The number of mitochondria and the mitochondria/cytosol ratio in cancer cells (Shapovalov et al, 2011) are comparable between all samples (Fig 4C). In contrast, the mean area of mitochondria was larger in cultured cells than in tumors and VR tumors that displayed smaller and denser mitochondria than R tumors (Fig 4D). Moreover, we observed qualitative

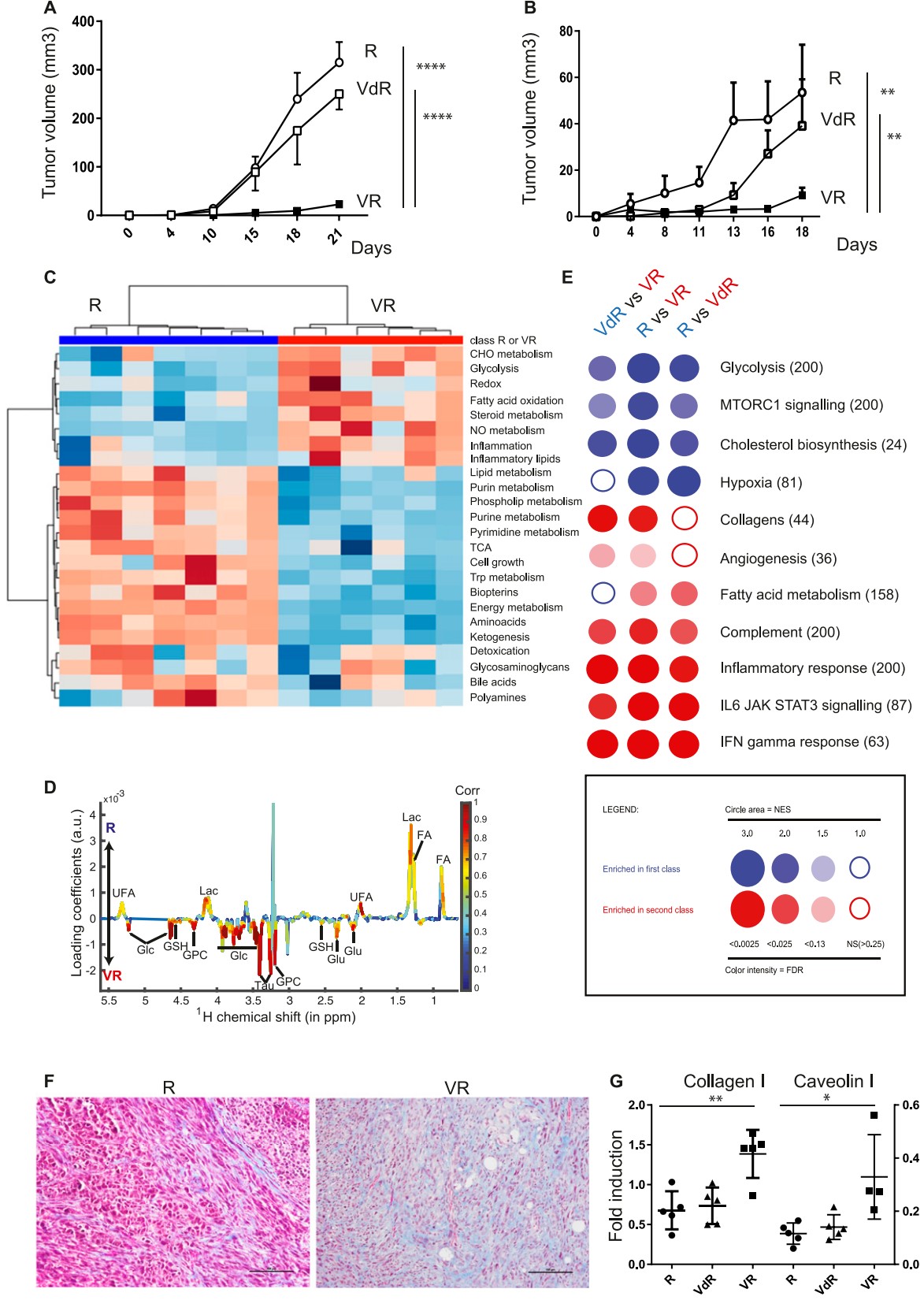

differences between R and VR tumors, which were not detected in the corresponding cultured cells (Figs 4E and F, and S6A and B). In VR tumors, the mitochondrial network is homogenous, dense, and frequently in contact with an organized ER. In sharp contrast, R tumors are less electron dense (reduced protein content) and show numerous features of mitochondrial stress with swollen organelles or presence of electron-translucent deposits, probably because of the presence of calcium phosphate precipitates (Giang et al, 2013) and reduction of mitochondrial crests (Shapovalov et al, 2011) and ER/mitochondria contacts associated with dilated ER (Bustos et al, 2017).

To evaluate mitochondrial activity, we compared basal and maximal oxygen consumption rate (OCR) between R and VR cell lines (Fig 4G). We found that VR cells showed increased basal and maximal respiratory potential and ATP production compared with R cells and by combining OCR and extracellular acidification rate (ECAR) measurements showed the profile of highly metabolic cells in vitro (Fig 4H), indicating the presence of a reservoir of functional mitochondria. Altogether, these results show that R and VR tumors do not show the same adaptation potential and that expression of Vnn1 presets cells in a metabolic state that allows them to maintain mitochondrial organization/activity in a tumor and likely stressed environment.

### Increasing pantetheinase activity in a Ras tumor limits its growth

In vivo, tumors are heterogeneous and may contain various proportions of Vnn1+ and Vnn1- cells. Furthermore, although Vnn1 expression levels are quite variable in tissues, one may wonder whether serum Vnn1 levels might be sufficient to compensate for the loss of Vnn1 expression in tumors (Rommelaere, Millet et al, 2013a). It has been previously shown that pantethine, the oxidized (disulfide) form of PantSH, can be converted in vivo into cysteamine and pantothenate (Wittwer et al, 1985) and has significant biological effects in infectious or inflammatory models (Penet et al, 2008; Kavian et al, 2015). We first verified that pharmacological doses of pantethine could reduce the growth of R tumors to a level comparable with that of a combination of cysteamine and pantothenate (Fig 5A). Next, we set out to determine whether a minimal amount of pantetheinase activity within the tumor mass would potentiate the antitumor effect and be able to control the growth of aggressive R tumors. To test this hypothesis, we injected a mix of R/VR tumor cells at a 10/1 cell ratio in immunocompetent C57BL/6 mice and, additionally, administered pantethine. Interestingly, the presence of 10% VR cells in an R tumor reduces tumor growth, and this inhibitory effect is further enhanced by the addition of pantethine

to mice (Fig 5B). Therefore, this experiment shows that a minimal amount of intra-tumor pantetheinase activity is required to generate a tumor suppressive context in aggressive tumors and suggests that the products of pantetheinase activity work in a paracrine mode on Vnn1- tumor cells.

## Discussion

Using Vnn1-deficient mice (Pitari et al, 2000), we have previously showed that pantetheinase activity is required for tissue adaptation to stress induced by damage, infection, or inflammation (Naquet et al, 2014) and more specifically during myofibroblast-dependent tissue repair and fibrosis (Dammanahalli et al, 2012; Kavian et al, 2016). In this study, we provide the first experimental evidence that induction of Vnn1 pantetheinase activity in a mouse sarcoma model contributes to the regeneration of mitochondrial CoA stores through the production of pantothenate from PantSH, leading to enhanced mitochondrial fitness. Indeed, although R and VR cell lines show comparable mitochondrial organization, growth, and differentiation potential when grown in culture, we observed a disorganization of the mitochondrial/ER network in R but not VR tumors in vivo. These results show that the hostile environment of the tumor decompensates mitochondrial homeostasis if its CoA stores are not restored by local pantetheinase activity. Moreover, in vitro–grown VR lines display a higher respiratory potential than R cell lines, in agreement with their ability to cope with metabolic stress. These results suggest that pantothenate availability determines mitochondrial fitness (thereby supporting ATP production) and that it is required for the fulfillment of effector functions observed in differentiated Vnn1+ tumors.

The results of this and several previous studies on the interplay between Vnn1 pantetheinase activity, CoA biosynthesis/degradation, and energy metabolism can be summarized schematically, providing a descriptive model that highlights the connections between the various systems (Fig 6). De novo CoA biosynthesis is initiated by the phosphorylation of pantothenate by pantothenate kinase (PanK), of which several isoforms with different subcellular locations and regulatory characteristics exist (Alfonso-Pecchio et al, 2012; Subramanian et al, 2016). All human PanKs experience feedback inhibition by acetyl-CoA and other acyl-CoAs, especially long-chain acyl-CoAs such as palmitoyl-CoA (Subramanian et al, 2016). This would suggest that PanKs would be inhibited when fatty acid pools are high. In the context of the current study, this correlates with the finding that fatty acids tend to

---

**Figure 2. Characterization of R and VR tumors grafted in mice.**
**(A, B)** Comparative growth curve of R, VdR, and VR J2A tumors (n = 7, t test and ANOVA, ***P < 0.001) in nude (A) and immunocompetent C57BL/6 (B) mice. **(C)** Metabolomics analysis of R and VR H1 tumors by LC-MS (n = 6) showing the hierarchical cluster of metabolic pathways according to cell types. Scaling is in unit of variance (mean/squared root of SD). **(D)** Metabolomics analysis of R and VR J2A tumors by NMR (n = 7) displaying the relative enrichment in metabolites in R (top panel) versus VR (bottom panel) cell lines. Orthogonal Projections to Latent Structures Discriminant Analysis loading plot are color-coded according to the correlation coefficients of each signal with the predictive component of the tumor samples. Metabolites with positive intensities are in higher concentrations in R samples, whereas metabolites with negative intensities are in higher concentrations in VR samples (P value of 0.008). **(E)** Transcriptomic analysis of J2A tumors (n = 5). BubbleGUM analysis displaying enrichment of gene sets on the pairwise comparisons of R, VdR, and VR tumors (NES, normalized enrichment score). The gene sets were downloaded from MSigDB (Broad Institute). The bubble area is proportional to the normalized enrichments score (NES) calculated by GSEA. The intensity of the color corresponds to the statistical significance of the enrichment. **(F, G)** Typing of the differentiation grade of J2A tumors by in situ collagen (Masson trichrome) versus hematoxylin staining (F) or qRT–PCR analysis of collagen 1 and caveolin 1 transcripts (n = 5, unpaired t test, *P < 0.05; **P < 0.01) (G). Lac, lactate; Glc, glucose; GSH, glutathione; Tau, taurine; GPC, glycerophocholine; FA, fatty acids; and UFA, unsaturated FA.

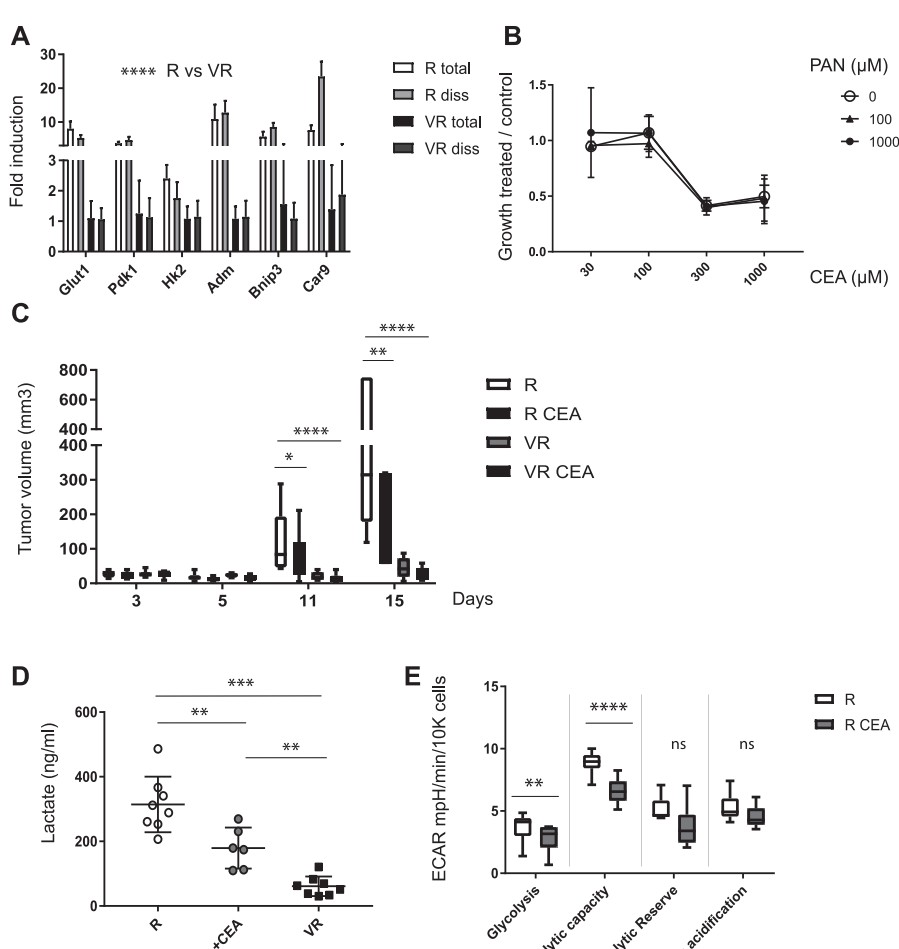

**Figure 3. Analysis of the Warburg signature.**
**(A)** Quantification by qRT–PCR of transcripts associated with the glycolytic/hypoxic signatures and tested on RNA from total H1 tumor mass (total, n = 3) or from dissociated (diss, n = 4) tumor cells after exclusion of the infiltrating CD45+ population by magnetic bead cell sorting (ANOVA statistics $P < 0.0001$). **(B)** Dosage of lactate production in H1 R and VR tumor samples (n = 6–8; $t$ test) from untreated or cysteamine (CEA)-treated mice when indicated. **(C)** Seahorse analysis of the in vitro–grown H1 R cell lines quantifying the effect of 500 $\mu$M cysteamine (CEA) extemporaneously added to the cell culture before ECAR measurement (ANOVA statistics $P < 0.0001$). **(D)** Effect of the addition of increasing concentrations of cysteamine (CEA) and pantothenate (PAN) to H1 cell cultures in vitro evaluated by crystal violet staining of cells (treated day 1 – control day 0)/(control day 1 – control day 0). **(E)** Growth of H1 tumors in mice treated or not with CEA ip (120 mg/kg) (n = 6, multiple $t$ test).

accumulate in the fast-growing R cell lines that rather rely on glucose for energy (Fig 2) and which also showed lowered CoA levels (Fig 4).

In contrast, the VR cells were found to accumulate glucose (Fig 2). Importantly, the expression of the cytosolic PanK4 isoform can be induced by high levels of glucose in a rat model (Li et al, 2005). PanK4 has no actual PanK activity (because of point mutations of key active site residues) but was recently shown to have a DUF89 phosphatase domain that acts on 4′-phosphopantetheine (PPantSH) to release PantSH, the substrate of Vnn1 pantetheinase (Huang et al, 2016). This would ensure supply of PantSH, which would have to exit the cell by an as yet uncharacterized mechanism, for Vnn1 to act on. The Vnn1-produced pantothenate and cysteamine would then enter the cell again, with the latter causing the observed inhibition of glycolysis and lactate formation (Fig 3) that would further enhance the factors leading to the accumulation of glucose. The pantothenate would feed the CoA biosynthetic pathway in the mitochondria, which is both distinct from and regulated differently from the one operating in the cytosol. Particularly, the mitochondrial PanK2 isoform is activated by palmitoyl carnitine (and likely also other carnitines) (Leonardi et al, 2007), counteracting the inhibition by acyl-CoAs (Fig 6). Although the levels of acyl-carnitine levels were comparable between the various cell lines in this study (not shown), the lower fatty acid levels and increased oxygen consumption of the VR cell lines is indicative of active $\beta$-oxidation occurring. Taken together, this would support our conclusion that the expression of Vnn1 leads to a metabolic reorganization that improves mitochondrial fitness through the increase of CoA in mitochondrial stores.

An important question that remains relates to the source of the PPantSH that serves as substrate for the PanK4-DUF89 phosphatase, which then indirectly feeds the extracellular Vnn1 pantetheinase. Although the complexity and interdependence of the various systems make the unequivocal determination of the source of the Vnn1 substrate experimentally challenging, our current knowledge suggest several possibilities. The pool of intracellular PPantSH is produced by several reactions: biosynthetically by the action of phosphopantothenoylcysteine synthetase (PPCS) and decarboxylase (PPCDC) on 4′-phosphopantothenate (PPan), the product of PanK, or through degradation of CoA by Nudix hydrolases (Strauss, 2010) (Fig 6). However, both these sources are unlikely, as using PPantSH in this manner would result in a futile cycle of biosynthesis and degradation.

Recent findings have suggested that the cytosolic and mitochondrial CoA pools are formed and maintained distinctly

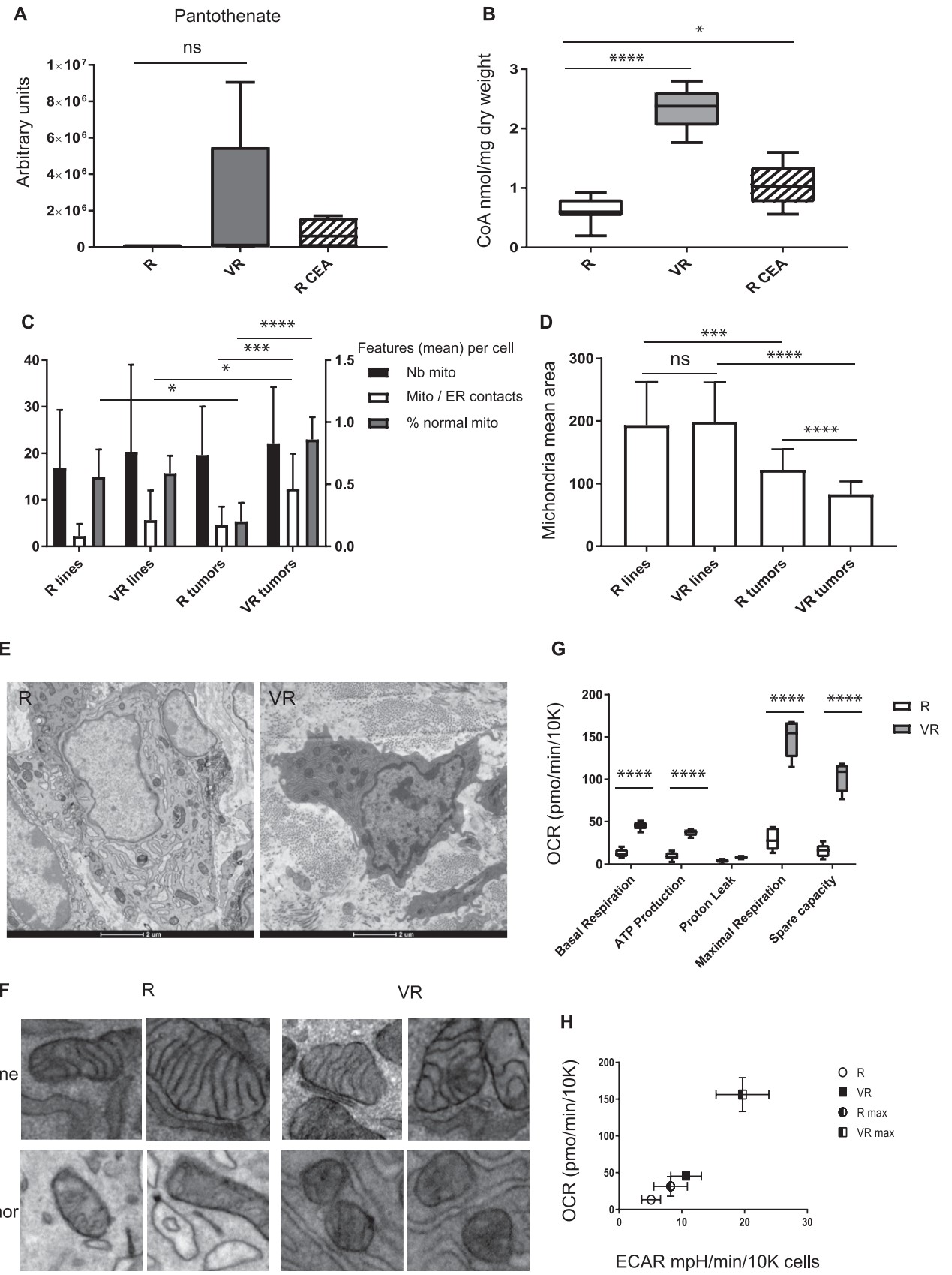

(Pietrocola et al, 2015). Specifically, mitochondrial CoA is formed from PPantSH by the CoA synthase (CoASy)—a bifunctional protein with both phosphopantetheine adenylyltransferase and dephospho-CoA kinase (DPCK) activities—that has been shown by two independent studies to be localized in the mitochondrial matrix (Rhee et al, 2013; Dusi et al, 2014). In contrast, CoA in the cytosol of Drosophila has been proposed to be derived from a separate predicted monofunctional DPCK (Vozza et al, 2017) acting on mitochondrial dephospho-CoA (dPCoA) exported by the SLC25A42 transporter (Fiermonte et al, 2009) (Fig 6). Although it is not yet certain if the latter also occurs in humans and the proposed strict subcellular compartmentalization of the CoA pools needs further experimental verification, the model as shown in Fig 6 would provide a mechanism whereby cytosolic PPantSH is able to feed Vnn1, while still allowing for mitochondrial CoA pools to be maintained.

However, it is more likely that the PPantSH that indirectly supplies Vnn1 is obtained from extracellular sources. Specifically, endonucleotide pyrophosphatases degrade extracellular CoA to form PPantSH, whereas there is some evidence that it is also excreted by certain bacteria (Jackowski & Rock, 1984). Coupled with the recent finding that extracellular PPantSH is relatively stable and is able to cross cell membranes in an unassisted manner (Srinivasan et al, 2015), this suggests that the microbiome could be a significant source of PPantSH. Such a conclusion would be in agreement with recent studies that highlighted the metabolic consequences of the close association of tumor and microbial cells (Bullman et al, 2017; Geller et al, 2017).

The cysteamine released by Vnn1 pantetheinase limits glycolysis and lactate formation, antagonizing the Warburg effect and thereby limiting cell growth while enhancing tumor cell differentiation and innate immune cell activation (Fig 2). It has long been known that sufficient ATP levels antagonize the rate-limiting phosphofructokinase-dependent step of glycolysis. Cysteamine, in redox equilibrium with cystamine (its disulfide), has been proposed to regulate biological functions through formation of mixed disulfides with enzymes (Martin et al, 2004; Naquet et al, 2014) or amino acids favoring the conversion of cystine into cysteine in cells (Elmonem et al, 2016). More specifically, it was shown using recombinant proteins that the catalytic activity of four glycolytic enzymes (hexokinase, phosphofructokinase, glucose-3-phosphate dehydrogenase, and pyruvate kinase) was reversibly inhibited by cystamine thiolation, whereas the same process activated the neoglucogenic 1,6-bisphosphatase (Terada et al, 1994). Taken together, these results suggest that cystamine could moderate glycolysis while favoring neoglucogenesis, synergizing with the ATP-mediated inhibition of glycolysis and further contributing to the reduction of the Warburg effect (Lu et al, 2015). Indirect regulation, for example, through the effect of cysteamine on transcriptional regulators, should therefore also be studied.

Taken together, the combination of the cysteamine and pantothenate-mediated effects is able to explain the rewiring imposed by Vnn1 expression on tumors, a process that cannot be recapitulated by the individual metabolites. In the STS model, lack of Vnn1 leads to the development of undifferentiated aggressive glycolytic tumors. These metabolic transitions are controlled by local cues and oncogenic drivers. It was shown that titrating the amount of Kras copy number defines metabolic reprogramming and therapeutic susceptibility (Kerr et al, 2016). This suggests that tumor cells calibrate their response to quantitative variations of cell-intrinsic or external factors and remain susceptible to modulation. Antagonizing the Warburg effect is, therefore, an attractive strategy to limit tumor growth, as documented by the impact of the metastasis suppressor KISS1 gene that enhances mitochondrial biogenesis and OXPHOS (Chen, 2012; Liu et al, 2013). Here, we demonstrate that this can be achieved by regulating the level of pantetheinase activity in tumors, therefore representing a new strategy to skew the tumor metabolic profile and limit its growth potential.

# Materials and Methods

### Mice

The *Ink4a/Arf* (p16/p19)-deficient mice were obtained from AM Schmitt-Verhulst (Huijbers et al, 2006). Vnn1$^{-/-}$, p16/19$^{-/-}$, and p16/p19/Vnn1$^{-/-}$ C57BL/6 mice were backcrossed for more than 15 generations on the C57BL/6 mouse background, genotyped for p16, p19, and Vnn1 deficiency as described (Pitari et al, 2000; Huijbers et al, 2006), fed on normal chow diet and maintained under specific pathogen-free conditions at the CIML facility. Experiments were performed on female and male mice and in accordance with institutional guidelines for animal care and use. This experimental design was authorized by the Ethical Committee for Animal Experimentation (no. 02820.03) (APAFIS 5875-2015061516136757-V3). The nude mice were maintained by Janvier, and experiments were carried out on female mice.

### Retroviral vectors

Human H-RasV12 fused in the N terminus with GFP (GFP/H-rasV12) inserted downstream of the intronic ribosomal entry site (IRES) sequence of the pQXCIX retroviral vector (Clontech) using the In-Fusion kit (Clontech). The mouse Vnn1 or catalytically deficient Vnn1dCr (Rommelaere et al, 2013a) sequences were inserted in the multiple cloning site downstream of the cytomegalovirus promoter. Three vectors described in Fig S1B were generated: pQCXIX-GFP/H-RasV12 (RasV12 or R), pQCXIX-Vnn1-IRES-GFP/H-RasV12 (Vnn1 RasV12

**Figure 4. Mitochondrial homeostasis.**
**(A)** Quantification of pantothenate levels by LC-MS analysis and (B) CoA levels in I1 tumor extracts (n = 9 for R and VR samples, n = 6 for CEA-treated R samples, unpaired *t* test). **(C–F)** EM analysis of mitochondria on individual cells observed in cultured grown cell lines and after grafting in nude mice (n = 20 cells/sample). **(C)** Quantification of mitochondria/cell in cell lines (H1 R/VR lines) and tumor extracts (H1 R/VR tumors) and quantification of mitochondrial features: contact points between ER and mitochondria (expressed in contacts per cell) and proportion of normal mitochondria per cell (i.e., organized crests, homogenous density) expressed as a ratio: normal/total mitochondria/cell. **(D)** Quantification of the mean surface are occupied by mitochondria expressed as a ratio: mitochondria surface area/number of mitochondria/cytosol. **(E, F)** Representative images of cells from R and VR tumors ([E] 8,500×), mitochondria in lines versus tumors (F), and two enlarged photos of 1 $\mu m^2$ per category 22,000×; see more in Fig S6. **(G)** Comparative analysis of the OCR for J2A R versus VR cultured cell lines (n = 6–7) by Seahorse analysis (ANOVA statistics *P* < 0.0001). **(H)** OCR versus ECAR plot for R and VR lines under basal or Carbonyl cyanide-4-(trifluoromethoxy)phenylhydrazone (FCCP)-treated conditions.

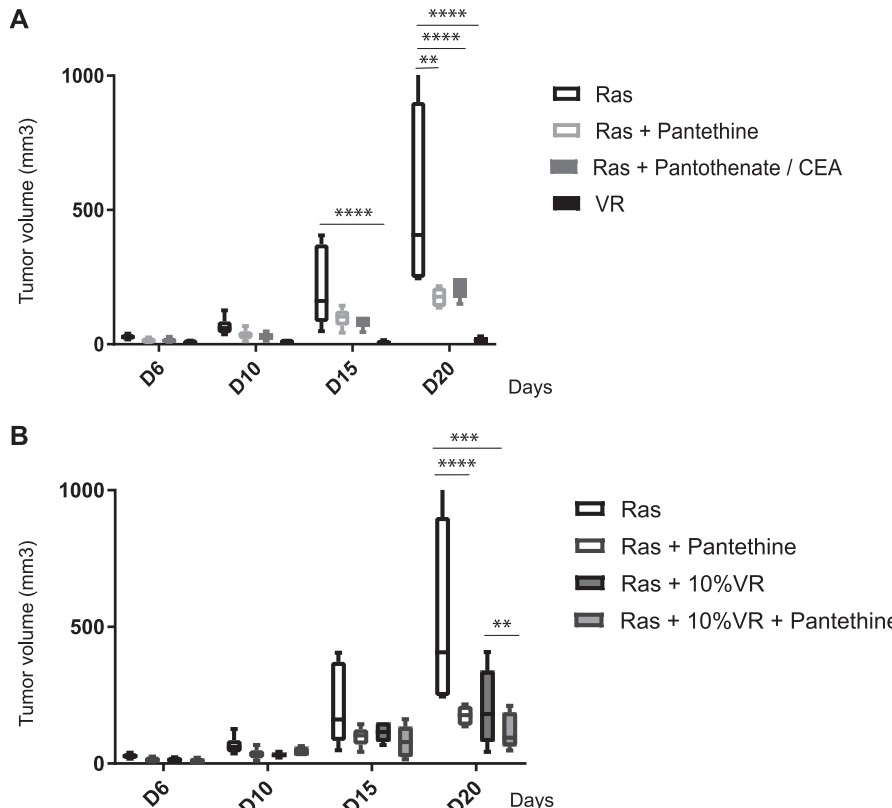

**Figure 5. Effect of increasing pantetheinase activity in R tumors.**
**(A)** Scoring of tumor growth in C57BL/6 mice grafted with R or VR cells and receiving pantethine or a mix of cysteamine and pantothenate every other day during the course of tumor development (n = 6–8 tumors per condition). **(B)** Comparative growth of R or R/VR chimeric tumors at two different cell ratios grafted in untreated or pantethine-treated mice. Multiple t tests were performed for statistical analysis.

or VR), and pQCXIX-Vnn1dCr-IRES-GFP/H-RasV12 (Vnn1dCrRasV12 or VdR). p16/p19/Vnn1$^{-/-}$ cells were then infected with retroviral particles obtained by the transfection of φNXe cells (American Tissue Culture Collection) with lipofectamine (Life Technologies)-containing retroviral vectors. Viral supernatants were collected 2 d later. For transformation, p16/p19 KO cells were seeded in 6-well plates and infected by a double 2 h-spinoculation with 8 μg/ml polybrene (Sigma-Aldrich).

### Pantetheinase activity

Pantetheinase activity was measured with the pantothenate-7-amino-4-methylcoumarin substrate (Ruan et al, 2010). Cultured cells (4 × 10$^5$) were lysed in 100 μl of PBS, 0.1% deoxycholate in the presence of protease inhibitor (Roche). Tissues were disrupted and lysed with a homogenizer in the same solution. After centrifugation at 10,000 g for 10 min, total protein concentration was measured in the supernatants using the BCA reagent (Pierce Thermo Scientific). Pantetheinase activity was measured by incubating 20–50 μg of total proteins in a final volume of 200 μl phosphate buffer (pH 8) containing 0.01% BSA, 1% DMSO, and 0.0025% Brij 35, 500 μM DTT for 10 min at RT before addition of 20 μM pantothenate-7-amino-4-methylcoumarin. The appearance of AMC was followed during the first 60 min of the reaction by scoring fluorescent signals at 355 nM using a fluorimeter (Tecan) and the slope corresponding to the production rate of the product of the reaction directly reflects the level of enzymatic activity.

### Cells

Aortic fibroblasts from p16/p19/Vnn1$^{-/-}$ mice were generated as described (Ray et al, 2001). Briefly, cells were isolated by enzymatic digestion with Collagenase P at 2 mg/ml (Roche) in DMEMF12 for 3 h and cultured in DMEMF12 supplemented with 10% FBS, 100 units/ml penicillin, 100 μg/ml streptomycin, 2 mM of L-glutamine, and 1 mM sodium pyruvate. We derived several myofibroblast cell lines from p16p19Vnn1$^{-/-}$ mice which upon transformation with an oncogenic Ras and transplantation in mice, grew as sarcomas within a few weeks. These lines called J2A, I1, and H1 were used indifferently in most experiments. For the quantification of population doubling time, 3 × 10$^5$ cells were seeded at low density in 12-well plate. Every 3 d, the cells were harvested with trypsin and counted with a cell counter (CASY). The curve shows the absolute cell numbers obtained on day 6, 12, and 18 of the culture, reset at 3 × 10$^5$ cells to the number of seeded cells (cell population fold increase). A batch of vials frozen at early passages was used for experiments to avoid tumor drift due to genetic instability and variant selection. Furthermore, the cells were systematically checked by flow cytometry to score RasGFP and Vnn1 levels and used when these levels were of comparable intensity (Fig S1C).

### Tumor grafts

2 × 10$^6$ transformed cells with plasmids containing RasV12 (R) or Vnn1 RasV12 (VR) or Vnn1 deficient for catalytically active

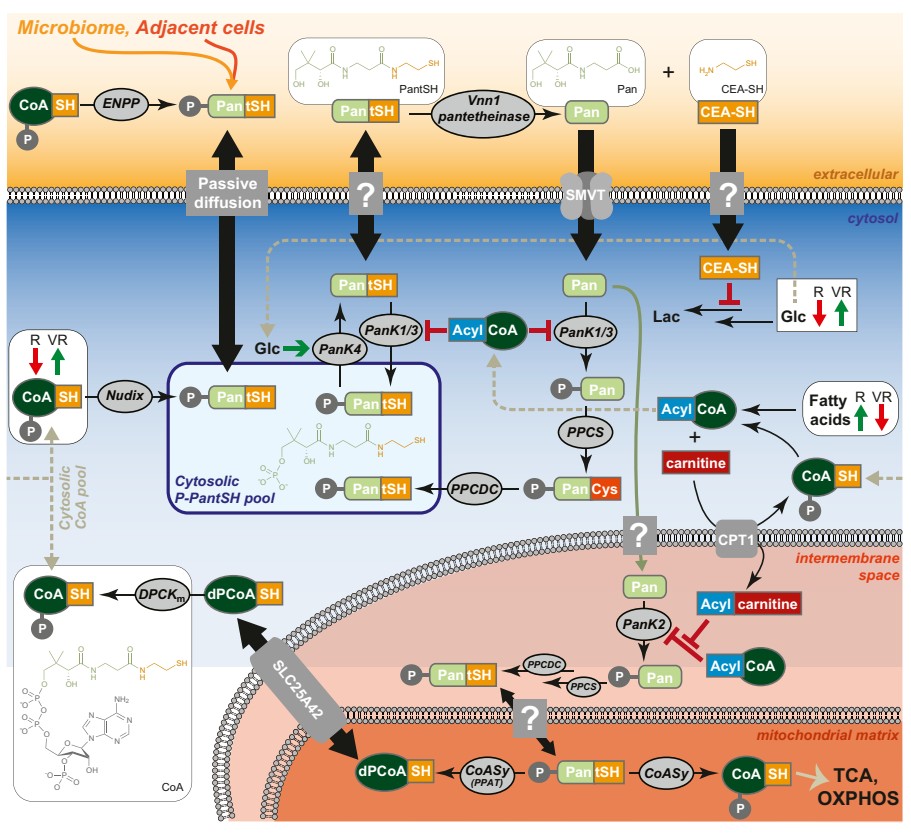

**Figure 6. Model of Vnn1-induced metabolic rewiring.**
The Vnn1 pantetheinase induced in VR cell lines catalyzes the hydrolysis of PantSH to form pantothenate (Pan) and cysteamine (CEA-SH). The Pan enters the cell through the solute multivitamin transporter (SMVT) and is transformed into CoA by the CoA biosynthetic machinery: PanK1β or PanK3, PPCS, PPCDC, and a predicted monofunctional DPCK (DPCK$_m$) in the cytosol, and PanK2, PPCS, PPCDC, and the bifunctional CoASy in the mitochondria (the subcellular location of PPCS and PPCDC remains to be confirmed, but independent studies have confirmed that both PanK2 and CoASy localizes to the mitochondria, the latter specifically to the matrix). The SLC25A42 transporter is predicted to link the cytosolic and mitochondrial dPCoA pools; however, its role in intracellular CoA and dPCoA transport is not yet fully understood. In R cell lines, fatty acid levels are increased, leading to the inhibition of the PanKs by acyl-CoAs and a reduction in CoA levels. In VR cells, fatty acid levels are lower, suggesting that they are shuttled to the mitochondria via the carnitine shuttle, of which the first component is carnitine palmitoyltransferase I (CPT1) that forms acylcarnitine in the intermembrane space, where it counteracts the inhibition of PanK2 by acyl-CoAs. The CEA-SH inhibits glycolysis and lactate (Lac) formation, leading to increased glucose (Glc) levels in VR cells; this induces the PanK4-DUF89 phosphatase that dephosphorylates P-PantSH and provides the PantSH used as substrate by Vnn1. The various possible sources of the cytosolic P-PantSH pool are shown and include cytosolic and mitochondrial biosynthesis, degradation by Nudix hydrolases and endonucleotide pyrophosphatases, and production from extracellular sources. Unknown or undefined transport mechanisms are indicated by gray boxes labeled with question marks. Abbreviations not defined in the legend are defined in the main text.

pantetheinase (dCr) RasV12 (VdR) were subcutaneously injected in the back of anesthetized (MiniTAG apparatus; Tem Sega) nude mice in 100 μl of PBS. C57BL/6 mice were irradiated at 4 Gy (RS2000 X-Ray irradiator) to transiently deplete peripheral lymphocytes and facilitate engraftment 1 d later. For chimeric R/VR grafts, $2 \times 10^6$ R cells was injected either alone or complemented with VR cells at a 10% VR/R ratio, checked by flow cytometry before injection. When indicated, pharmacological compounds (Sigma-Aldrich) were administered intraperitoneally 2 d after grafting, and then every 2 d at a dose of 120 mg/kg for cysteamine HCl, 500 mg/kg for pantothenate, and 1 g/kg for pantethine. Tumor growth was quantified on anesthetized mice by measuring the length (L) and width (W) of the tumor mass and extrapolating its volume using the following formula: $(L \times W^2)/2$. A precise quantification was systematically performed at the time of harvest on dissected tumors. Mice were sacrificed when tumor size reached a maximum of around 400 mm$^3$. The statistical analysis of tumor growth was performed using a two-way ANOVA or multiple $t$ tests.

## Tumor dissociation

Cell suspensions were prepared using the Miltenyi Biotech tumor dissociation kit. Briefly, the tumor tissue was enzymatically digested in a GentleMACS Octo Dissociator. The samples were filtered through a Cell Strainer (Becton Dickinson) to remove cell clumps.

Immunocytes were depleted using an AutoMACS separator after labeling with mouse CD45 MicroBeads. The samples were analyzed on an LSR II UV or Canto II cytometer (BD Biosciences) equipped with a Diva software.

## Anatomopathology and histochemistry

All tumors were fixed in 10% buffered formalin (pH 7.4) and paraffin-embedded. The sections were analyzed blindly by a veterinary anatomopathologist. Histological grading was performed as published (Coindre, 2006) taking into account the histological scoring (1: sarcomas closely resembling normal adult mesenchymal tissue; 2: sarcomas for which histologic typing is certain; and 3: embryonal and undifferentiated sarcomas, sarcomas of doubtful type), mitotic count (1: 0–9 mitoses; 2: 10–19 mitoses; and 3: >20 mitoses per 10 high power field), and necrosis score (0: none; 1: <50% necrosis; and 2: >50 necrosis). The final grade is defined by adding scores (grade 1: total score 2, 3; grade 2: total score 4, 5; and grade 3: total score 6, 7, 8). For collagen quantification, tissues were stained with Masson Trichrome (blue/green) and counterstained with hematoxylin (purple). For immunohistochemistry staining of mitochondria, paraformaldehyde-fixed sections were subjected to heat-induced epitope retrieval in citrate buffer (pH 6) by boiling at 98°C in a water bath for 20 min, and subsequent cooling to RT for 20 min. Briefly, the endogenous peroxidase activity was blocked

with 3% $H_2O_2$ in distilled water for 10 min. Then, the sections were incubated for 1 h at RT with a rabbit anti-Tom20 Ab diluted at 1/100. Excess primary Ab was removed three times by washing in PBS for 5 min before incubating with a specific anti-rabbit HRP Ab (Vector kit impress HRP reagent kit ref: MP-7401) at RT for 30 min. The secondary Ab was washed three times in PBS for 5 min before the DAB solution (Vector: peroxidase substrate kit DAB ref: SK-4100) was applied to the sections and incubated at RT for 5 min. The sections were counterstained with Mayer's hematoxylin and mounted with Entellan and examined under microscope (Nikon).

### Flow cytometry

For induction of Vnn1 expression by myofibroblast cell lines, 1 $\mu$g poly(I:C) was transfected using lipofectamin, whereas thapsigargin (50 nM) and antimycin A (50 $\mu$M) for ER or mitochondrial stress inducers, respectively, were added to the cultures 24 h before flow cytometry analysis (Canto II; Becton Dickinson) using the 407 anti-Vnn1 mAb (Aurrand-Lions et al, 1996) coupled to iFluor 633 (ReadiLink Antibody Labeling kit; AAT Bioquest). Expression of H-RasV12 was controlled by quantifying the level of GFP fluorescence (for H-RasV12).

### Electron microscopy analysis

Samples were prepared using the NCMIR protocol for serial block-face scanning electron microscopy (West et al, 2010). 70-nm ultrathin sections were performed on a Leica UCT Ultramicrotome (Leica) and deposited on formvar-coated slot grids. The grids were observed in an FEI Tecnai G2 at 200 KeV, and acquisition was performed on a Veleta camera (Olympus). Quantification of mitochondrial size was performed by measuring with Image J the area occupied by all mitochondria divided by the number of mitochondria identified in each cytosol. This experiment was performed for R and VR cell lines and tumors (n = 20 cells per condition).

### qRT–PCR analyses and transcriptomic analysis

Total mRNA from tissues or cells was purified using the RNeasy Mini Kit (Qiagen). For qRT–PCR analysis, 0.5–1 $\mu$g RNA was reverse transcribed with the SuperScript II RT kit (Life Technologies). Amplification was performed on a 7500 Fast Real Time PCR system (Applied Biosystems) using SYBR green Master Mix (Takara) and specific primer pairs (Table S1). Expression levels were normalized to the control gene hypoxanthine phosphoribosyl transferase. Statistical analysis of gene expression was performed with a $t$ or ANOVA test. For transcriptomic analysis, mRNAs were extracted from four independent R, VdR, or VR J2A tumor samples harvested on day 21, quality checked using a Bioanalyser (Agilent) and sent to the "Plate-Forme Biopuces et séquençage" of Institut de Génétique et de Biologie Moléculaire et Cellulaire (Illkirch) for analysis using affy_mogene_1_0_st_v1 arrays. Raw and robust multichip average-normalized data have been deposited on GEO database under reference GSE107686. Enrichment analyses based on pairwise comparisons were performed using the BubbleGUM software (http://www.ciml.univ-mrs.fr/applications/BubbleGUM/index.html) which permits to perform high-throughput GSEA (Spinelli et al, 2015).

### Seahorse analysis

ECAR and OCR were measured using a Seahorse XF-24 Metabolic Flux Analyzer. Myofibroblasts cell lines were seeded at $2 \times 10^4$ cells/well on Seahorse XF24 V7 multi-well plates for 16 h at 37°C in 10% $CO_2$. For the ECAR assay, 1 h before measurement, cell culture medium was replaced with DMEM (0 mM glucose) supplemented with 2 mM glutamine and incubated at 37°C in a non-$CO_2$ incubator. Cysteamine (500 $\mu$M) was added as indicated at this phase of treatment. ECAR was measured under basal conditions and after sequential addition of 10 mM glucose, 1 $\mu$M oligomycin, and 100 mM of 2-deoxyglucose. To record mitochondrial activity, the same assay medium was used supplemented with 1 mM sodium pyruvate and 10 mM glucose. OCR was measured before and after sequential injections of 1 $\mu$M oligomycin, 1 $\mu$M (carbonyl cyanide 4-(trifluoromethoxy)phenylhydrazone) FCCP, and 0.5 $\mu$M of antimycin A/rotenone. ECAR and OCR data were normalized to cell numbers using a cell titration curve quantified by crystal violet staining.

### LC-MS and NMR-based metabolomics

For both analyses, technical details for metabolite extraction are provided in supplementary information. Annotation was performed by matching exact mass to publicly available databases and from plausible biological context. The resulting matrix was curated from unannotated peaks and used for further statistical analysis. Metabolites identified by LC-MS analysis were grouped into metabolic function performed on the ground of literature data (Tables S2 and S3).

#### *LC-MS*
Frozen tumor tissue samples were lyophilized during 20 h in a PowerDry LL3000 Freeze Drier. 10 mg of dried tissue was milled in a fine powder in a FastPrep-24 homogenizer. Metabolites were extracted in 1.5 ml of a mixture of ice-cold acetonitrile (ACN):milliQ water (1:1). The samples were vortexed for 30 s, sonicated for 10 min at 4°C in a Diagenode's Bioruptor Pico, and centrifuged at 10,000 $g$ at 4°C for 5 min. Supernatants were stored at −80°C until evaporation for analysis. Metabolomic analysis was performed on a Dionex 3000 LC system, hyphenated through an ESI source to a Brüker Impact QTOF mass spectrometer.

The analyses were performed sequentially, both in positive and negative ionization mode to extend metabolome coverage. HILIC LC-MS analysis was performed using a Merck polymeric bead–based ZIC-pHILIC column (150 mm × 2.1 mm × 5 $\mu$m, 2A) with a gradient elution of ACN containing 0.1% formic acid and water containing 16 mM ammonium formate at 250 $\mu$l/min flow rate and a column temperature of 25°C. The injection volume was 5 $\mu$l. The LC conditions were as follows: solvent A—16 mM ammonium formate in water, solvent B—0.1% formic acid in ACN, solvent B 97% for 2 min, then B to 70% in 8 min, then B to 10% in 5 min, held 2 min, then back to B 97% in 1 min, and held 4 min, vials tray 4°C. For the MS settings, the capillary voltage was set at 3,500 V (positive mode), and the nebulizing parameters were set as follows: nebulizing gas ($N_2$) pressure at 50.8 psi, drying gas ($N_2$) flow at 12 liters/min, and drying temperature at 200°C. Mass spectra were recorded from $m/z$ 50 to 1,200. After acquisition, raw data files were converted into mzXML

before or to XCMS processing to generate a data matrix of deconvoluted ions with their intensity, m/z, and retention times (Martin et al, 2015). LC-MS analytical drift was corrected using the method described by Van der Kloet (van der Kloet et al, 2009). The LC-MS metabolomics profiling identified 91 metabolites which were sorted into discrete pathways (Table S2) based on data in the literature, and a numerical score was calculated for each pathway by multi-bloc analysis (Table S3). Each block was then analyzed as a separate variable. Pantothenate measurement was at the limit of detection in our LC-MS analysis preventing a robust statistical analysis.

### NMR

For each sample, about 15 mg of thawed tissue was placed into a 30-ml disposable insert in which 10 ml of $D_2O$ was added. The disposable insert was then introduced into a 4-mm $ZrO_2$ HRMAS rotor before NMR analysis. The NMR experiments were recorded on a Bruker Avance III spectrometer equipped with a $^1H/^{13}C/^{31}P$ HRMAS probe and operating at 400 MHz and 100 MHz for $^1H$ and $^{13}C$, respectively. All the spectra were recorded at a spinning rate of 4,000 Hz and a temperature of 277 K. For each sample, the $^1H$ HRMAS NMR spectrum was acquired using the Carr-Purcell-Meiboom-Gill NMR sequence, preceded by a water presaturation pulse during relaxation time of 2 s. This sequence enabled us to reduce the macromolecule and lipid signal intensities using a T2 filter of 50 ms synchronized with the rotor rotation speed. For each spectrum, 128 free induction decays of 16,384 data points were collected using 8,000 Hz of spectral window. The free induction decays were multiplied by an exponential weighting function corresponding to a line broadening of 0.3 Hz and zero-filled before Fourier transformation. The calibration on the alanine doublet ($\delta$ = 1.47 ppm) and the phase and baseline corrections and the alignment of shifted signals were performed using NMRprocflow online software (Jacob et al, 2017). The identification of the NMR signals was carried out using $^1H$-$^1H$ TOCSY, $^1H$-$^{13}C$ HSQC NMR experiments, online databases (Human Metabolome Database [Wishart et al, 2013]) and literature (Nicholson et al, 1995). Each spectrum was divided into 0.005-ppm-width integrated buckets and normalized to total spectrum intensity after removing the pre-saturated water signal region (4.85–5.10 ppm) using NMRprocflow software to create the dataset matrix. The water region must be removed because of inconsistent signal suppression. The data matrix was then exported to the software Simca-P v14 (Umetrics) for statistical analysis.

### Lactate quantification

Lactate concentrations were quantified according to the manufacturer's protocols (Sigma-Aldrich). Briefly, tissues were homogenized in four volumes of lactate assay buffer and centrifuged at 13,000 $g$ for 10 min to remove insoluble material. Samples were deproteinized with a 10-kD molecular weight cut-off spin filter. A master reaction mix containing 46 $\mu$l lactate assay buffer, 2 $\mu$l lactate enzyme mix, and 2 $\mu$l lactate probe was added on 50 $\mu$l sample solution. Reactions were incubated at RT for 30 min and sample absorbance measured at 570 nm on a microplate reader.

### CoA quantification

Quantification of CoA was performed as described (Goosen & Strauss, 2017). Tissue extracts were prepared following a modification of a protocol described (Siudeja et al, 2011). Tumors were lyophilized for 20 h and then homogenized. 10 mg of dry tissue were resuspended, mixed, and vortexed in 300 $\mu$l of ice-cold extraction solution consisting of 50% ACN and 50% water. Then 450 $\mu$l of 100% ACN and 37.5 $\mu$l of 2 M formic acid was added to obtain a 80% ACN 0.1 M formic acid final solution. The samples were sonicated for 10 min at 4°C (10 cycles of 30 s ON/30 s OFF) to complete tissue lysis and metabolite extraction. Cell debris were removed by centrifugation at 10,000 $g$ for 15 min at 4°C. The supernatant (about 700 $\mu$l) was then filtered by centrifugation through Nanosep 3K membrane filtration tubes for 10 min at 10,000 $g$. 223 $\mu$l of the filtered extract was neutralized with 70 $\mu$l Tris buffer (500 mM, pH 7.6) and 328.1 $\mu$l of water. The disulfides were reduced by addition of 47.3 $\mu$l of 1 mM tris(2-carboxyethyl)phosphine. After 10 min, 31.5 $\mu$l of $N$-[4-(7-diethylamino-4-methyl-3-coumarinyl)phenyl]maleimide (10 mM in 100% ACN) was added to bring the total volume to 700 $\mu$l. The samples were subsequently lyophilized before shipment for analysis; upon arrival, lyophilized samples were reconstituted in 50% aqueous ACN before HPLC analysis.

### Biological resources are structured by the clinical departments in various databases

Conticabase tumor bank (conticabase.org) for soft tissue and visceral sarcoma includes n = 15,808 samples with clinical annotations and follow-up (https://conticabase.sarcomabcb.org/). The analysis of gene expression was conducted on selected samples of this database; data and methods are accessible online on atg-sarc.sarcomabcb.org.

### Statistical analysis

Analysis was performed on the GraphPad Prism 7.03 software. Values are represented with standard deviations or minimum to maximum plots. We used ANOVA, and $t$ and log rank tests for tumor size evaluation and survival curves, respectively. For metabolomics analysis, partial least squares-discriminant analysis, a supervised statistical method that attempts to separate the three groups of samples by regressing on a so-called dummy y-vector consisting of the class codes, was used for classification purposes. All experiments performed are recapitulated in Table S4.

## Supplementary Information

## Acknowledgements

C Giessner and T Gensollen were supported by a doctoral contract from Aix Marseille University. Financial resources originated from institutional

funding from CNRS, INSERM, AMU, and Cancéropôle Provence Alpes Cote d'Azur (PACA) (2014). This work was initiated through a collaborative Institut National du Cancer (INCA) grant (2008 - 045) and benefited from a funding Equipes Fondation pour la Recherche Médicale 2014 (DEQ20140329532) for specific experiments. E Strauss was supported by a grant from the South African National Research Foundation. JY Blay was supported by funds from NetSARC, LYRIC (INCA-DGOS 4664), LYon Recherche Innovation contre le CANCer, European Clinical trials in Rare Sarcomas (FP7-278742), and European network for Rare Adult solid Cancer. B Dieme, L Shintu, and JC Martin were supported by a grant from Cancéropôle PACA. The electron microscopy experiments were performed on PiCSL-FBI core facility (IBDM, AMU-Marseille), member of the France-BioImaging national research infrastructure. We thank T Bouriaud for her technical help; people involved at the CIML imaging, flow cytometry, histology, and animal facilities; and Alice Carrier and Laurence Borge for assistance with the use of the Cell Culture Platform facility (Centre de Recherche contre le Cancer de Marseille, U1068) in charge of the SeaHorse platform. We thank B Malissen, P Golstein, Lee Leserman, and AM Schmitt-Verhulst for critical reading of the manuscript. The authors are indebted to Pr Silvestro Dupré who pioneered the work on pantetheinase.

## Author Contributions

C Giessner: data curation, formal analysis, investigation, methodology, and project administration.

V Millet: data curation, formal analysis, investigation, and methodology.

KJ Mostert: methodology.

T Gensollen: data curation and methodology.

T-P Vu Manh: formal analysis, investigation, and methodology.

M Garibal: methodology.

B Dieme: methodology.

N Attaf-Bouabdallah: methodology.

L Chasson: methodology.

N Brouilly: formal analysis and methodology.

C Laprie: formal analysis, investigation, and methodology.

T Lesluyes: methodology.

J-Y Blay: validation, investigation, and methodology.

L Shintu: formal analysis and methodology.

JC Martin: investigation and methodology.

E Strauss: conceptualization, funding acquisition, validation, methodology, and writing—original draft, review, and editing.

F Galland: conceptualization, supervision, and methodology.

P Naquet: conceptualization, formal analysis, supervision, funding acquisition, validation, investigation, methodology, project administration, and writing—original draft, review, and editing

## Conflict of Interest Statement

The authors declare that they have no conflict of interest.

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
