## [Reviewer comments · Life Science Alliance]

Vnn1 pantetheinase limits the Warburg effect and sarcoma growth by rescuing mitochondrial activity

Philippe Naquet, Erick Strauss, Franck Galland, Caroline Giessner, Virginie Millet, Konrad Mostert, Thomas Gensollen, Thien-Phong VU MANH, Marc Garibal, Binta Dieme, Noudjoud Attaf-Bouabdallah, Lionel Chasson, Nicolas Brouilly, Caroline Laprie, Tom Lesluyes, Jean-Yves Blay, Laetitia Shintu, and Jean Charles Martin
DOI: 10.26508/lsa.201800073

Review timeline:

Submission Date:	23 March 2018
Editorial Decision:	4 May 2018
Revision Received:	25 June 2018
Editorial Decision:	9 July 2018
Revision Received:	12 July 2018
Accepted:	13 July 2018

Report:

(Note: Letters and reports are not edited. The original formatting of letters and referee reports may not be reflected in this compilation.)

May 4, 2018

Re: Life Science Alliance manuscript #LSA-2018-00073-T

Prof. Philippe Naquet
Centre d'Immunologie de Marseille Luminy
INSERM-CNRS-Univ. Méditerranée Case 906 Cedex 9
Marseille 13288
France

Dear Dr. Naquet,

Thank you for submitting your manuscript entitled "Vnn1 pantetheinase rescues coenzyme A-dependent mitochondrial activity in sarcomas" to Life Science Alliance. The manuscript was assessed by expert reviewers, whose comments are appended to this letter. We invite you to submit a revision if you can address the reviewers' key concerns, as outlined here.

As you will see, all three reviewers appreciate your data and support publication of your work in Life Science Alliance. A few issues are raised, but they seem straightforward to address, most of them by adding clarifications in the text and by considering improved data representation. The additional control mentioned by reviewer #2 (GFP-RasV12 expression levels being indeed equal in all three cell lines used) should be added. The new LC-MS/MS analyses suggested by reviewer #3 is not mandatory for acceptance here.

-- High-resolution figure, supplementary figure and video files uploaded as individual files: See our detailed guidelines for preparing your production-ready images, <http://life-science-alliance.org/authorguide>

-- Summary blurb (enter in submission system): A short text summarizing in a single sentence the study (max. 200 characters including spaces). This text is used in conjunction with the titles of papers, hence should be informative and complementary to the title and running title. It should

describe the context and significance of the findings for a general readership; it should be written in the present tense and refer to the work in the third person. Author names should not be mentioned.

B. MANUSCRIPT ORGANIZATION AND FORMATTING:

Full guidelines are available on our Instructions for Authors page, <http://life-science-alliance.org/authorguide>

Thank you for this interesting contribution to Life Science Alliance. We are looking forward to receiving your revised manuscript.

Sincerely,

Reviewer #1 (Comments to the Authors (Required)):

Manuscript by Giessnet et al.

The authors show a series of experiments investigating differences between various cells and

models expressing or not expressing Vnn leading to interesting findings. Experiments are well controlled, and various analysis and techniques are performed using a large variety of methods, such as histology, activity assays, metabolomics, transcriptomics, EM and more.

The manuscript contains some twists which are difficult to follow, some experiments and the rationale behind should be better explained and introduced. Some conclusions are not justified. These are further explained below.

In general the authors should improve the clarity of the manuscript, it contains different fields and reading and understanding all the figures is hard, the figure legends in general are very minimal, some figures are a bit cryptic, e.g for a person familiar with EM it is not possible what is presented and important information in figure 2D. Some figures show bad quality, but this could have happened during the conversion. The discussion is quite lengthy and the presented figure and model interesting but complex. Figure legends for the supplementary figures are missing, therefore those are difficult to read and interpret.

Specific comments:

What is not clear to me is the title, claiming the coenzyme A-dependency, this is not demonstrated convincingly.

First sentence of the summary, omit the word "of"

The authors claim that induction of the Vnn1 pathway rescues mitochondrial activity, however, not the pathway is investigated, but it is merely expression of Vnn1.

The authors show that "In the only case of STS (III) observed in the Vnn^{+/+} mice, a low level of Vnn1 expression was confirmed by qRT-PCR. What exactly do the authors mean by confirmed?, confirm what? What are the levels of other grade tumors, does expression of Vnn in general correlate with differentiation of tumors, meaning less expression in the type II and more in type 1 in the Vnn^{+/+} background.

Figure 2B, what are the histological preparations visualized, what staining? What are the colors? Addition of a zoom in and a zoom out will be more informative instead of a rather large picture containing no extra information. Some more information is required here.

Figure S1A, Vnn1 is expressed in Vnn1^{-/-} ??????

The authors show that in the metabolomics analysis R segregates from VR tumors, what about the VdR, does it segregate also from VR but not from R?

Figure 2 D is not clear,

How do I read this figure? What is representing the R and what the VR? This figure is also not well presented, could have happened in the uploading procedure of the manuscript by the authors.

Figure 2F, what is the collagen staining? More colors are visible, it is not clear what type of staining was used here.

Figure 3B explain under 3B what CEA means. Present the results in order, now it is explained in 3C, but presented earlier in 2B.

Better introduce the usage of cysteamine in the experiments, why was this tested? To compensate for the absence of Vnn? Is there less cysteamine present in R lines compared to VR lines? What happens in case cysteamine and pantothenate are added together?

CoA levels are increased in VR tumors compared to R tumors, what about VdR tumors? Is this only in the specific cells and conditions as presented in the manuscript. Upon overexpression of Vnn or downregulation of Vnn in other cells, are there differences in CoA levels. Is there a difference in CoA levels in other tissue of the p16/p19^{-/-} mice versus the p16/p19/Vnn^{-/-} mice?

Regarding the EM analysis, what do the authors mean by an organized ER. Is there a difference in size of the mitochondria? What do the authors mean by numerous features of mitochondrial stress? Which features, they only list 4 features. Is this seen in all mitochondria? Are the 4 features present in the same mitochondria?

What do the authors mean by the last sentence in a stressed environment (last sentence of the result section)?

In the discussion the authors claim that induction of Vnn1 pant activity contributes to regeneration of mitochondrial CoA stores through the production of pantothenate from PantSH, but this is an assumption and a possible route of events explaining their observations. They do show that CoA levels in cells are influenced by the presence or absence of Vnn1, however, whether this is via the route they state is not clear.

On page 10, in the middle, why is here the Dusi reference mentioned? Please check other references as well carefully.

The discussion suggests an interesting and complex interplay of various pathways.

The figure does not show how CoA is synthesized in the cytosol, the last step is missing. COASY is also found in the cytoplasm (at least this cannot be excluded), why is this step only shown to take place in the mitochondria?

Why is there a cross on the membrane where CoA enters the cell? Because this does not occur? Then why visualizing this?

Why are CoA levels increased in the VR lines compared to the R lines? This is not well explained in the model.

Reviewer #2 (Comments to the Authors (Required)):

Based on previous work on Vnn1 Giessner et al demonstrate a tumour-suppressive function of the pantetheinase enzyme, in soft tissue sarcoma (STS). They then develop a novel STS model to investigate the role of Vnn1 in metabolism in this context. In basic terms Vnn1 expression is able to combat the Warburg effect by promoting improved mitochondrial metabolism in STS cells. This is an interesting and well-constructed study and broadly a well-written manuscript. However, more explanation of the reasoning for performing each experiment would be beneficial.

Would the authors agree that a more physiologically relevant model would have been to extract cells from the skin STS tumours from the p19/p16^{-/-} Vnn1^{-/-} and p19/p16^{-/-} Vnn1^{+/+} mice to use in the xenograft experiments?

It is likely that these would have not required ectopic RasV12 expression as they were transformed in vivo. Also, comparing cells from the Vnn1^{-/-} and Vnn1^{+/+} mice directly would have removed the requirement for forced Vnn1 expression. It should be explained in the text why this model was not

selected.

Could the authors please explain in the text the rationale for using oncogenic Ras to transform their cell lines rather than any other oncogene?

It is critical to ensure that the levels of Ras expression are equal across the three cell lines. A difference in Ras expression alone could account for the growth, metabolic and gene expression differences. Although the authors have shown flow-cytometry data of these cells, due to the critical nature of these controls, the authors should also present a graph displaying total GFP-RasV12 expression (from the flow-cytometry data) as well as a western blot comparing GFP-RasV12 expression across all cell lines used.

The authors should also present more clearly that the expression of Vnn1 in the VR and VdR cell lines is comparable. This could take the form of a graph showing total Vnn1 expression from the flow cytometry data and/or a western blot.

The authors state that addition of pantothenate does not alter the glycolytic capacity of R cell line in vitro. Does pantothenate alter cell proliferation or expression of hypoxia genes, as CEA does?

The authors suggest that Vnn1 pantetheinase activity results in the increased CoA levels observed in VR tumours. What are the CoA levels in VdR tumours? In the text it is suggested that increased CoA in VR tumours is due to the production of pantothenate (a CoA precursor) by Vnn1. However, the authors show in this figure that CEA treatment increases CoA levels in R tumours. What do the authors believe is the significance of this?

Some other issues that require attention:

- Labelling error on figure S1A? Showing increased Vnn1 expression in Vnn1^{-/-} cells?
- Remove "Lorem ipsum" text from middle of figure S2
- Figure S2E is very confusing and requires simplification
- Figure S1E - should show number of cells over time, not passage.
- Define the relevant abbreviations in the figure legends (VdR, R, VR etc)

Reviewer #3 (Comments to the Authors (Required)):

The authors present a well-written and structured report of the effect of deletion of the Vnn1 pantetheinase gene in a murine model of cancer. The manuscript is improved by both mechanistic investigations and translationally relevant correlations in a patient population. Controls including the catalytic dead VdR model are especially well done, and all conclusions reached are supported with a reasonable degree of confidence for publication. Criticisms are minor, but certain experimental elements are weak and the manuscript may actually be improved by removing them due to problems interpreting their results.

Major criticism

1. The authors should point to data or references that the Vnn1 gene deletion in a similar mouse background is not deleterious for survival since major conclusions of the manuscript make that assumption.
2. The untargeted LC-MS analysis is questionable. As written in the methods, there is insufficient description of techniques to assess rigor, but assuming the techniques are relatively standard along

with the description given (simple water/organic extraction, no isotope labeled amino acids used as surrogate internal standards, HILIC analysis in full scan mode at Qtof resolution of 17.5-35k) the data from these experiments only detracts from the manuscript. The methods description for the LC-MS alludes to more details provided in supplemental, but I do not see any more details (other than uninformative tables of peak area for questionably specific assignments of molecular identity). As a sum of NMR, MS and transcriptomic data I have no concerns about the conclusions- but the LC-MS data is problematic in isolation (see minor comment 6 below for one example). I think it would strengthen the manuscript to remove the "specific" molecules presented in SFig 3/Table S2 (perhaps with exception of pantothenate- see minor criticisms 6 and 8).

3. If the authors have time and residual tissues from the mice, or can culture the fibroblasts-targeted isotope dilution LC-MS/MS based profiling of pantothenate, pantetheine, cysteamine, phosph-pantothenate with a validated assay would add considerable weight to the manuscript. This would be expected to take ~ a month and would lend credence to the model as presented. However, it does require specialized techniques and instrumentation that the authors may not have access to quickly.

Minor Criticisms

1. Fig 1. Making the color scheme colorblind friendly for panel B would be appreciated. The powering of panel C is unclear. Panel D is at a resolution that makes the text in the figure impossible to read.
2. Page 5 First paragraph, the statement "these results suggest that Vnn1 can be expressed in STS and may delay their development" could be better stated as ... "in STS and may have protect against of limit cancer growth." The data as presented do appear to discriminate between inhibition at different steps of carcinogenesis and cancer growth. Additionally, a citation should be provided for the Conticabase data.
3. Page 5 Second paragraph, missing a closed () in the clause of transfected poly(I:C)...
4. Outside the scope of this manuscript, the observation that it required Ras to allow tumor development even in Nude mouse xenografts in interesting. Considering the cell of origin (versus the potential cancer progenitors in the first mouse model) this may warrant future work. It could indicate, for example, a differentiation block dependent on CoA metabolism at some point of carcinogenesis. Considering the implications for acylation of histones based on the CoA data presented this may be fruitful.
5. Fig 2. Panel A- unclear what test is being applied where (legend says t-test and ANOVA, without specification or notation). Panel C- scale is missing.
6. SFig 3. Given the methods presented, I cannot access the reliability of the metabolite measurements (especially regarding specificity). It is unclear if isomers are resolved (presenting isocitrate), and certainly not likely that 15(S)-HETE was specifically measured given the R enantiomer is also biologically present and there are a number of isomers including 5-, 12-, and 11- with distinct biological roles. Given that this data does not really alter the nature or other findings of the study it remains a minor criticism.
7. Fig 3. B. How was lactate production assessed? Can the authors confirm that cysteamine at 500 microM does not interfere with the SeaHorse media buffering?
8. Lack of difference in pantothenate is hard to interpret without knowing variability in the LC-MS assay used. Likewise, how was phospho-pantothenate accounted for?
9. How do these results compare to the Vnn1 ablation in Sf-1 mice? See (Latre de Late P, El Wakil A, Jarjat M, de Krijger RR, Heckert LL, Naquet P, Lalli E. Vanin-1 inactivation antagonizes the development of adrenocortical neoplasia in Sf-1 transgenic mice. *Endocrinology*. 2014 Jul;155(7):2349-54. doi: 10.1210/en.2014-1088. Epub 2014 Apr 8. PubMed PMID: 24712878.

Answers to reviewer 1

We addressed all the points raised by the reviewer

Claiming CoA dependency in the title:

We agree with the referee that this point has not been formally demonstrated and we can only correlate the expression of Vnn1 to the recycling of the pantothenate / CoA axis and then to mitochondrial metabolism. There is no easy way to formally demonstrate this point in vivo in the absence of a dynamic analysis of the metabolic pathway using radiolabeled metabolites, a method to which we have no access and which cannot be used in our institute due to the absence of a legal authorization to administer radiolabeled compounds to mice. We therefore rephrased the title and some of the conclusions made in the summary.

Sarcoma models: p16p19^{-/-} STS versus transformed myofibroblast cell lines

As explained in the first paragraph of the result section and shown in Table 1, STS were rare in p16p19^{-/-} mice and only one case of skin fibrosarcomas was found out of 30 mice coming from 3 independent cohorts kept until the age of 200 days. The other STS observed in this mouse genetic background were found in the spleen/liver where other cells expressing Vnn1 can be found, therefore qRT-PCR is not informative. Therefore, we could only perform a qRT-PCR analysis on the single skin STS found in these mice. As indicated in the text, this sarcoma was a grade III STS and qRT-PCR analysis of this tumor allowed the detection of Vnn1 transcripts although to a low level as expected for an aggressive tumor. It has been previously shown that Vnn1 can be expressed by myofibroblasts in vivo (references Dammanahalli et al and Kaviani et al) and our own results using myofibroblast cell lines (Figure S1) show that Vnn1 is expressed by these cells in vitro and that its expression is further induced by various intracellular stresses. Our interpretation of these results is what was proposed in our manuscript: Vnn1 transcripts are detectable in this skin fibrosarcomas but, in agreement with the high grade of this tumor, Vnn1 levels are very low, probably due to the emergence of fast growing Vnn1- variants from the initial tumor, a situation that we have already observed on a few occasions. In other tumors from Vnn1^{+/+} mice which were not STS, Vnn1 expression levels mainly reflected the level in the tissue where the tumor was identified (i.e. liver, spleen, ...) and irrelevant to the tumor itself. New pictures were taken and shown at two magnifications (x200 and x400, revised Figure 1).

Concerning the grade I to III STS growing in the p16p19Vnn1^{-/-} mice, no Vnn1 transcripts could be detected as expected.

In Figure S1A, there was an inversion in the Figure between Vnn1^{-/-} and Vnn1^{+/+} tumors. This has been corrected.

NMR analysis:

In Figure 2D, S2C and S2D, the results of the statistical analysis of OPLSDA loading plots are color-coded according to the correlation coefficients of each NMR signal with the predictive component of the tumor samples (the figure legend have been rephrased). Metabolites with positive intensities are in higher concentrations in R samples whereas metabolites with negative intensities are in higher concentrations in VR samples (p-value of 0.008) (Figure 2D). As shown in Figures S2C and S2D which compare R / VdR and VdR / VR samples respectively, there is no difference between R and VdR (Figure S2c: p-value of 0.33) whereas VdR and VR segregate (Figure S2d: p-value of 0.04) and showed the expected enrichment in fatty acids (FA) in VdR versus glucose (Glc) in VR tumors as with R tumors. Given the workload and cost of the LC-MS analysis, this was not performed for the VdR tumors.

In Figure 2F, the staining protocol has been clarified in the Methods section (Masson trichrome + hematoxyline).

Legends of Figures 2 and 3 have been expanded.

Cysteamine administration:

Our working model suggests that the amount of pantetheinase metabolites becomes limiting in an aggressive tumor and therefore adding back either the enzyme and/or its substrate/products should compensate for this partial deficiency and exert a paracrine control on R tumors.

We changed the order of the text and figure that describe the glycolytic signature to better introduce the rationale for testing cysteamine and pantothenate in our experiments. As shown in Fig 3B, in vitro, addition of cysteamine reduced cell growth by 50% but pantothenate (at different concentrations) did not further modify this result.

To further strengthen the importance of pantetheinase metabolites in the control of tumor growth, we performed a novel experiment and added these results to Fig 5. We injected a mix of R and VR tumor cells ($R \gg VR$) and further complemented mice with the substrate (in a well-tolerated reduced form called pantethine) or the products (cysteamine and pantothenate) of pantetheinase activity. The objective was to demonstrate that the presence of a pantetheinase activity in a heterogeneous tumor, containing Vnn1+ and Vnn1- cells, a naturally occurring situation in vivo, is sufficient to generate a tumor suppressive context even for R tumors. Interestingly, the presence of only 10% VR cells in a R tumor reduces tumor growth by 50% and this inhibitory effect is further increased by the addition of pantethine to mice.

Since cysteamine had an impact on the growth and glycolytic signature of R tumors, we wished to test cysteamine impact on CoA levels. As shown in Figure 4B, although statistically significant, this effect might not be biologically relevant and could be an indirect consequence of the growth inhibitory effect or the difference in the tumor microenvironment.

Quantification of CoA levels:

VdR tumors: We recapitulated in the supplementary Table S5 all the experiments using various cell lines and experimental protocols. In initial experiments concerning tumor growth potential, transcriptomic, qRT-PCR, NMR and Sea Horse analyses, we systematically compared R, VdR and VR tumors. Results of growth curves shown in Figure 2 and S2 (independent experiments on various cell lines) indicate that VdR tumors grow faster than VR tumors and to a level identical or slightly reduced compared to R tumors. VdR tumors displayed undifferentiated and hypoxic signatures (Figures 2E, 2G, S2) and metabolomics analysis by NMR showed that they had a R-like phenotype.

Furthermore, we cannot formally exclude the possibility that VdR tumors might display a residual pantetheinase activity towards the physiological substrate pantetheine in vivo. Indeed, the VdR molecule contains the non-catalytic base domain shown by Boersma et al to regulate enzymatic activity and to have putative partner proteins. A residual activity might not be detected using the chemical derivative pAMC tested as a substrate for in vitro quantification of pantetheinase activity but might explain some experimental variability when comparing R and VdR tumors (Figure S2). Given this uncontrollable risk and the cost of other experiments, we used mostly R and VR tumors for LC-MS, CoA quantification and EM analysis and in addition tested in most cases the impact of cysteamine administration to mice.

CoA concentrations in the p16p19 model: In earlier experiments using Vnn1 deficient mice, we did not have access to the recently published technique for CoA analysis developed by E Strauss's team which provides a higher sensitivity than all the available kits currently on the market. So this analysis was not performed in the p16p19 model. In addition, as already explained, the rare occurrence of fibrosarcomas in p16p19^{-/-} mice would have prevented a robust appreciation of this parameter.

Concerning the EM analysis, we evaluated the average surface area occupied by mitochondria by quantifying surface areas with the Image J software and dividing the total mitochondrial area by the number of mitochondria observed in each cytosol (n= 20 per condition). Results are added to Figure 4D and show that mitochondria are smaller in tumors than cell lines, but also smaller in VR versus R tumors. These results are discussed in the text.

Text: We slightly modified the last sentence of the results section by saying: "Altogether, these results show that R and VR tumors do not show the same adaptation potential and that expression of Vnn1 presets cells in a metabolic state that allows them to maintain mitochondrial organization/activity "in a tumor and likely stressed" environment".

Model:

We attempted to integrate our results in the current knowledge on CoA metabolism. The contribution of phosphopantetheine stems from the recent paper from O Sibon's group which suggests that this compound is much more stable than pantetheine in vivo, detectable in the serum and therefore susceptible to diffuse in a tumor.

Concerning the localization of COASY, three recent papers (all cited including Dusi's) converge towards the notion that this enzyme is only detectable in the mitochondria. Combined with the discovery of a dephospho-CoA transporter and a cytosolic monofunctional dephospho-CoA kinase (DPCK) in fruit flies, this suggests the model for the synthesis of CoA as presented. We have nonetheless made this information clearer in the legend of the figure.

Extract from the Dusi's reference:

"Several studies have investigated the subcellular compartmentalization of the CoA biosynthetic pathway and have demonstrated that both PANK2, defective in the most common NBIA disorder, and CoA synthase alpha and beta are mitochondrial enzymes. PANK2 is mainly located in the intermembrane space (2,15,16) whereas CoA synthase alpha and beta are anchored to the outer mitochondrial membrane by the N-terminal region (17) or localized within the mitochondrial matrix (18). **We here demonstrate that COASY is mainly located in the mitochondrial matrix** and that the identified amino acid substitution causes instability of the protein with altered function of its enzymatic activity."

The increase in CoA levels in the VR model would be the direct consequence of enhanced regeneration of pantothenate and this has been better explained in the legend.

The Figure 6 presenting the model has been modified to take into account the comments of the reviewer.

Answers to reviewer 2

Tumor graft models:

In our initial experiments, we attempted to derive tumor lines from the primary tumors. However, two technical limitations appeared: first, the efficacy of derivation of cell lines was not optimal and second, we had obtained only one skin fibrosarcoma from p16p19^{-/-} mice. To stabilize the tumor cells and adapt them to in vitro growth conditions, we had to reinject dissociated cells from the primary tumor; then, we observed the emergence of tumors with a very high growth potential in vivo (this was the case for the J1 cell line not used further in our study). The genetic and cellular instability of these tumors was therefore the main limitations for a comparative usage in further phenotypic characterization.

We also derived primary myofibroblast cell lines from both p16p19^{-/-} and p16p19Vnn1^{-/-} mice (Figure S1A). In our experience with grafted p16p19^{-/-} tumors, we noticed that tumors tended to progressively lose Vnn1 expression in vivo, probably due to epigenetic modifications, and that variants with an aggressive behavior emerged. Therefore, we chose to use lines from Vnn1-deficient hosts using a transfected version of catalytically-active / dead Vnn1 pantetheinase activity where Vnn1 expression is stable in vivo.

The use of the Ras-V12 oncogene was based on previous experience in the laboratory and on published literature showing that Ras-driven oncogenesis rapidly leads to glycolytic tumors and that in sarcoma patients, Ras mutations are associated with tumors with poor prognosis. Since we had in mind to test the impact of Vnn1 transfection on the Warburg effect, we privileged this choice. However, we plan to extend this work using different models of sarcoma cell lines derived from mutagenesis-induced carcinogenesis. This is beyond the scope of this manuscript.

Ras and Vnn1 quantification in transformed cell lines:

Primary myofibroblast cell lines were transformed using similar procedures, sometimes several times during the course of the study. Indeed, we tried to limit the risk of emergence of variant cell lines by limiting their use for a limited number of in vitro passages.

We systematically quantified Ras-GFP versus Vnn1 (APC) expression by flow cytometry post transformation. Both GFP-Ras and Vnn1 levels moderately differed between cell lines and results obtained after transformation and prior to their use in various experiments are combined in Table infra which has been added as Table S1 in the paper. Concerning GFP Ras expression, whereas the H1 and J2A R cell lines express slightly higher levels than the corresponding VR line, the opposite is found for the I1 cell line. Despite these differences the growth of R / VdR lines is always higher than that of VR cell lines. Therefore, whereas Ras expression levels might have an impact on the growth rate in vivo (in general I1 > H1 > J2A, see Figure S2A), the presence of Vnn1 is associated with a slow growth potential.

Since the J2A cell type displayed homogeneous expression levels between R, VdR and VR cell lines and high levels of Vnn1 expression in VdR and VR cell lines, it served as a reference throughout the study and used in priority in all experimental set ups. The I1 and H1 cell lines were used in several additional experiments (see Table S5 which could be added to the manuscript is requested).

Added as Table S1: Cytofluorometric analysis of transformed cell line quantifying Ras GFP versus Vnn1 APC mean fluorescence intensity (MFI)

	Ras GFP MFI	Vnn1 MFI
--	-------------	----------

	H1	I1	J2A	H1	I1	J2A
Control	112	38	58	94	46	190
R	1919	690	2607	98	33	358
VR	1260	1850	2001	13603	27337	70242
VdR	1411	2100	2672	12382	23848	86713

The corresponding flow cytometric profiles are shown infra.

A western blot analysis was not performed in these initial experiments. To validate the flow cytometry results with a comparative western blot analysis, we prepared cell lysates from freshly thawed H1 cell lines in RIPA buffer and simultaneously performed a flow cytometric analysis. These cell samples are issued from later in vitro passages and not representative of those used in our initial experiments. After quantification using the BCA reagent, similar amount of protein was loaded per lane. Blots were revealed with an anti-GFP antibody, quantified and normalized using an actin antibody as a control tested on the same blot. As can be seen in panel C of the following figure, there is a perfect match between the western blot analysis and the flow cytometry analysis. This analysis retrospectively validates the flow cytometry method to compare various cell lines.

Pantothenate effect:

As shown in Fig 3B, in vitro, addition of pantothenate does not change growth of the cell lines in the absence or presence of cysteamine, which reduced cell growth by 50%. In vivo administration of pantothenate and cysteamine was performed to tumor bearing mice and results are presented seen in Fig 5A. However, it is difficult to precisely monitor how much pantothenate indeed reaches the tumor. To further strengthen the importance of pantetheinase metabolites in the control of tumor growth, we performed a novel experiment and added these results to Fig 5B. We injected a mix of R and VR tumor cells ($R \gg VR$) and further complemented mice with the substrate (in a well-tolerated reduced form called pantethine) or the products (cysteamine and pantothenate) of pantetheinase activity. The objective was to demonstrate that the presence of a pantetheinase activity in a heterogeneous tumor, containing Vnn1+ and Vnn1- cells, a naturally occurring situation in vivo, is sufficient to generate a tumor suppressive context even for R tumors. Interestingly, the presence of only 10% VR cells in a R tumor reduces tumor growth by 50% and this inhibitory effect is further increased by the addition of pantethine to mice.

Quantification of CoA levels:

VdR tumors: We recapitulated in the supplementary Table S5 all the experiments using various cell lines and experimental protocols. In initial experiments concerning tumor growth potential, transcriptomic, qRT-PCR, NMR and Sea Horse analyses, we systematically compared R, VdR and VR tumors. Results of growth curves shown in Figure 2 and S2 (independent experiments on various cell lines) indicate that VdR tumors grow faster than VR tumors and to a level identical or slightly reduced compared to R tumors. VdR tumors displayed undifferentiated and hypoxic signatures (Figures 2E, 2G, S2) and metabolomics analysis by NMR showed that they had a R-like phenotype.

Furthermore, we cannot formally exclude the possibility that VdR tumors might a residual pantetheinase activity towards the physiological substrate pantetheine in vivo. Indeed, the VdR

molecule contains the non-catalytic base domain shown by Boersma et al to regulate enzymatic activity and to have putative partner proteins. A residual activity might not be detected using the chemical derivative pAMC used as a substrate for in vitro quantification of pantetheinase activity but might explain some experimental variability when comparing R and VdR tumors (Figure S2). Given this uncontrollable risk and the cost of other experiments, we used mostly R and VR tumors for LC-MS, CoA quantification and EM analysis but tested in most cases the impact of cysteamine administration to mice.

Since cysteamine had an impact on the growth and glycolytic signature of R tumors, we wished to test cysteamine impact on CoA levels. As shown in Figure 4B, although statistically significant, this effect might not be biologically relevant and could be an indirect consequence of the growth inhibitory effect or the difference in the tumor microenvironment.

Other comments have been taken into account.

Answers to reviewer 3

Impact of Vnn1 gene deletion on mouse survival

We had previously scored mouse longevity (See Figure infra, % survival versus days) in the Vnn1^{-/-} versus control BALB/c background (n=20 mice / genotype) and the results show that Vnn1 deficiency does not significantly affect longevity (C57BL/6 mice were not specifically tested for this phenotype but we never suspected any abnormality in our mouse colony).

Untargeted LC-MS analysis

We did not consider as a main issue the detailed extraction technique, since as mentioned by the reviewer this is quite standard. In the method section, we detailed steps concerning sample extraction for LCMS data. Nonetheless, it should be noticed that our aim was not to provide a quantitative but rather a comparative information among the experimental conditions with regards to the putatively annotated metabolites. We performed a non-targeted metabolomics analysis, for which no reliable quantitative method currently exists. Using a single isotopically labelled reference cannot provide a reliable quantitation for all detected metabolites, due to the varied and uncontrolled ionization rate of the eluting metabolites. Quantitation requires a labelled standard for each metabolite, but since their identity is not known prior to the experiment, it is thus impossible to address this issue. Quantitation is only feasible when limited to a pre-defined list of metabolites to be analyzed, not when addressing untargeted analyses.

Also completing the experiment as suggested could be very challenging to us and not easily feasible within a reasonable time-frame. Moreover, starting a new cell culture batch at months later than those done for the present paper would distort the analysis, since even well-controlled experiments when reproduced are usually not fully stackable.

As suggested by the reviewer, we have excluded results presented in Figure S3 and Table S2 and included results for pantothenate quantification in Figure 4A.

Effect and quantification of Vnn1-related metabolites (cysteamine, pantothenate, coenzyme A)

To better justify the use of Vnn products in our experiments, we reformatted the text and the Figure 3. Cysteamine levels were not tested on cell lines *in vitro* or in tumors. In early work using Vnn1 deficient mice (Pitari et al 2000), we showed that cysteamine levels were undetectable in Vnn1 deficient mice compared to control mice. However, as discussed in another review (Naquet et al, 2014), cysteamine levels are difficult to quantify with precision *in vivo* due to the possible coupling of cysteamine to proteins, preventing an accurate quantification of free cysteamine in tissues.

Furthermore, mice express another pantetheinase isoform (Vnn3) with a differential tissue expression pattern, which can partially compensate for cysteamine deficiency *in vivo*. Therefore, evaluation of cysteamine levels *in vivo* is tricky and not so informative. Concerning pantothenate, we provide in Figure 4 the quantification performed from tumor extracts using LC-MS analysis. As discussed in the text, levels are quite low and at the limit of detection, leading to variability on the results (from undetectable to low), and preventing robust measurements. However, as shown in Figure 4, a majority of R tumors showed undetectable levels of pantothenate compared to VR tumors where levels were quite variable. Evaluation of phosphopantothenate levels are not usually performed in such analyses as its levels are expected to be even lower than that of pantothenate. Its accurate quantification would therefore require labeled standards as explained above.

Isotopic labelling

Our research institute is not any more accredited for the use of labelled isotopes and these techniques are unfortunately not available in our environment. We had previously contacted several metabolomic platforms for this specific question without success. Therefore, despite the obvious interest of this analysis, we are not able to perform directly or indirectly these experiments at this stage.

Minor criticisms

- 1 We changed the colors for panel B, and improved the resolution of panel D. Concerning panel C, this graph has no statistical value. We just provided the expression levels for collagen I and α SMA, markers of sarcoma differentiation, on the only available STS observed in this mouse model.
- 2 The sentence has taken in consideration the comment. Indeed, we have no argument in the p16/19 model that Vnn1 affects one or the other aspect of tumorigenesis. In contrast, in the cell line model, our analysis indicates that the growth and differentiation status of fully competent tumor cell lines is affected by Vnn1 expression. The Conticodatabase is referenced in the method section.
- 3 () corrected
- 4 This point is absolutely relevant. In the absence of an oncogenic Ras, the growth of cell lines is extremely low *in vivo*. We sometimes observed a small subcutaneous mass which failed to generate a tumor and usually spontaneously resolved with time. This occurred with Vnn1+ and Vnn1- myofibroblast cell lines. However, this analysis was not performed enough times to provide any statistical power. We have not explored the level of histone acylation in our tumors, neither quantified the acetyl CoA / CoA ratio. We can't exclude a contribution of these parameters to the observed phenotype.
- 5 Statistical analysis of data in Figure 2 was performed with t-tests for comparison of end points between two tumor types (shown in the Figure). ANOVA as also performed with the same result but not shown for this Figure (we therefore corrected this sentence. Concerning the LC-MS analysis, the scaling is in unit of variance (mean / squared root of standard deviation). It is now specified in figure legend.

For Figure S2A, we performed ANOVAs to examine independent graft experiments.

ANOVA table for Figure S2A: analysis of R versus VR tumors

Two-way ANOVA	Ordinary	Alpha	0,05	
Source of Variation	% of total variation	P value	P value summary	Significant?

Interaction	17,85	0,0019	**	Yes
Cell line	16,1	0,0031	**	Yes
Vnn1	38,79	<0,0001	****	Yes

ANOVA table	SS (Type III)	DF	MS	F (DFn, DFd)	P value
Interaction	199679	2	99840	F (2, 26) = 8,078	P=0,0019
Cell line	180090	2	90045	F (2, 26) = 7,285	P=0,0031
Vnn1	433793	1	433793	F (1, 26) = 35,1	P<0,0001
Residual	321350	26	12360		
Number of missing values		4			

- 6 We decided to leave out the results presented in Fig S3 which do not provide at this stage further information. But indeed the reviewer is right, we cannot precisely define the isomers unless chromatographic time differences are wide, and this is a limitation of such an approach. It is now specified in the text.
- 7 Lactate production was measured using a commercial kit from Sigma as indicated in the Mat and method section. Concerning cysteamine addition to the SeaHorse medium, there is no change in pH up to 1 mM cysteamine.
- 8 Pantothenate measurement was at the limit of detection in our LC-MS analysis preventing a robust statistical analysis and one would expect phosphopantothenate to be present at even lower concentrations. We were using a well-validated workflow (doi.org/10.1007/s11306-014-0740-0) for data post-processing and quality control. In fact, all ion features over 30% of coefficient of variation in the quality control samples (made up of a pooled aliquot of each sample) were discarded, so that the instrumental variation is kept well below the biological variation for each ion. This allow performing statistical comparison. Two independent experiments were tempted using different amounts of tissue. Many tumors scored negatively, mostly of the R genotype, preventing a clear interpretation of our results. An additional limitation to this analysis concerned VR tumors which are of very small size preventing the analysis of isolated tumors. Therefore, we had to pool 3-4 tumors from different animals to be able to detect a signal.
- 9 We were also puzzled by this result since in the SF-1 mouse model, Vnn1 overexpression favored the development of dysplastic lesions in the surrenal. The p16p19 model was set up to be able to explore the role of Vnn1 in various tumor models. First, the contribution of Vnn1 in the Latre's paper is part of a multi-partner molecular context. Indeed, the SF-1 transgene drives the overexpression of several target genes including Vnn1 but also regulators of redox status (GSTA). We previously found that the lack of Vnn1 affects the redox balance in irradiated or infected animals but this parameter was not explored in the surrenal model. Furthermore, in the sarcoma model, we did not detect major changes in the redox status of tumors. Therefore, the two models cannot be compared from a mechanistic point of view. Second, we did not explore the metabolic status of surrenal tumor cells since this model was not easy to experimentally manipulate and mostly depended on the spontaneous emergence of small tumors in vivo. Third, surrenal tumors derive from epithelial and not mesenchymal origin and this could also affect our conclusions. Few reports describing cancer-associated variations in VNN1 expression in patients have been reported but mostly considered VNN1 as a putative marker. Finally, even if one considers the link between Vnn1, mitochondrial metabolism and cancer development, there are still many debated issues on the role of FAO / OXPHOS in tumor growth versus metastatic progression. Therefore, although this question is of obvious interest, we have not yet found a global and simple scheme explaining all these observations, unfortunately.

July 9, 2018

RE: Life Science Alliance Manuscript #LSA-2018-00073-TR

Prof. Philippe Naquet
Centre d'Immunologie de Marseille Luminy
INSERM-CNRS-Univ. Méditerranée Case 906 Cedex 9
Marseille 13288
France

Dear Dr. Naquet,

Thank you for submitting your revised manuscript entitled "Vnn1 pantetheinase limits the Warburg effect and sarcoma growth by rescuing mitochondrial activity". As you will see, all reviewers support now publication, pending satisfactory minor revision.

We would thus like to invite you to address the remaining comments of reviewer #2 and to provide a final version of your manuscript. Please note that there is an issue with the mitochondria selected for magnification in Figure 4F and S6. Please make sure that your quantifications are correct and that all magnifications match the correct source data samples; currently a VR line mitochondrion is magnified as a R line one and vice versa. Please also add scale bars to Fig S5A and provide descriptions in the legends for all panels displayed in this figure.

A. FINAL FILES:

-- High-resolution figure, supplementary figure and video files uploaded as individual files: See our detailed guidelines for preparing your production-ready images, <http://life-science-alliance.org/authorguide>

B. MANUSCRIPT ORGANIZATION AND FORMATTING:

Full guidelines are available on our Instructions for Authors page, <http://life-science-alliance.org/authorguide>

Sincerely,

Reviewer #1 (Comments to the Authors (Required)):

The authors touch upon a complex field of CoA metabolism and how this influences growth of tumor cells. They reveal a strong influential role of Vnn1 pantetheinase on the Warburg effect using various models techniques and approaches. The authors addressed all my concerns and the

manuscript reads also in a much more comprehensive way. In the revised version, the data are supportive for the claims. There are no additional issues to be addressed.

Reviewer #2 (Comments to the Authors (Required)):

The authors have generally addressed my concerns.

I am happy with the use of flow cytometry to analyse Vnn1 and Ras expression in the cell lines. However, the current figures and Table S1 do not allow the reader to easily make a quantitative assessment of the potential differences in Ras/Vnn1 expression. To remedy this, the authors should show the flow cytometry data similarly to how it was presented in the rebuttal.

Figure S1E should show increase in cell number over a quantifiable time

Reviewer #3 (Comments to the Authors (Required)):

The authors sufficiently and quite reasonably answered all the queries to my satisfaction. I enjoyed their additions to the manuscript and look forward to seeing the article in print.

2nd Authors' Response to Reviewers: July 12, 2018

Answers to the editor, Reference LSA-2018-00073-T

Marseille july 12, 2018

Dear Andrea Leibfried,

We wish to submit the final version of our manuscript including the requested corrections. All modifications in the main text are written in red color.

We modified the Fig 4 to take into account the request to see different pictures of mitochondria than those presented in FigS6 and also to correct the correspondence and scales of images when needed. We added the flow cytometry data requested by reviewer 2 to Fig S1 and added the scale bar on Fig S5. Consequently, Table S1 disappeared as it is redundant with the data presented in this Fig S1. We also homogenized the legends and the text format in the new figures.

I did not know whether you still considered the information presented in Table S4 (recapitulation of experiment) in this final version. I included it but it could be removed in requested.

I hope these modifications will satisfy all these useful improvements and remain available if needed.

Thank you for your help in this process, yours sincerely

Philippe Naquet

July 13, 2018

RE: Life Science Alliance Manuscript #LSA-2018-00073-TRR

Prof. Philippe Naquet
Centre d'Immunologie de Marseille Luminy
INSERM-CNRS-Univ. Méditerranée Case 906 Cedex 9
Marseille 13288
France

Dear Dr. Naquet,

Thank you for submitting your Research Article entitled "Vnn1 pantetheinase limits the Warburg effect and sarcoma growth by rescuing mitochondrial activity". I appreciate the introduced changes, and it is a pleasure to let you know that your manuscript is now accepted for publication in Life Science Alliance. Congratulations on this interesting work.

The final published version of your manuscript will be deposited by us to PubMed Central (PMC) as soon as we are allowed to do so, the application for PMC indexing has been filed. You may be eligible to also deposit your Life Science Alliance article in PMC or PMC Europe yourself, which will then allow others to find out about your work by Pubmed searches right away. Such author-initiated deposition is possible/mandated for work funded by eg NIH, HHMI, ERC, MRC, Cancer Research UK, Teletthon, EMBL.

Please also see:

<https://www.ncbi.nlm.nih.gov/pmc/about/authorms/>

<https://europepmc.org/Help#howsubsmanu>

*****IMPORTANT:** If you will be unreachable at any time, please provide us with the email address of an alternate author. Failure to respond to routine queries may lead to unavoidable delays in publication.*******

DISTRIBUTION OF MATERIALS:

Again, congratulations on a very nice paper. I hope you found the review process to be constructive and are pleased with how the manuscript was handled editorially. We look forward to future exciting submissions from your lab.

Sincerely,
